# Soil moisture–atmosphere coupling accelerates global warming

Liang Qiao[1,5], Zhiyan Zuo[1,5] ✉, Renhe Zhang[1] ✉, Shilong Piao[2], Dong Xiao[3] & Kaiwen Zhang[1,4]

Soil moisture–atmosphere coupling (SA) amplifies greenhouse gas-driven global warming via changes in surface heat balance. The Scenario Model Intercomparison Project projects an acceleration in SA-driven warming due to the 'warmer climate – drier soil' feedback, which continuously warms the globe and thereby exerts an acceleration effect on global warming. The projection shows that SA-driven warming exceeds 0.5 °C over extratropical landmasses by the end of the 21st Century. The likelihood of extreme high temperatures will additionally increase by about 10% over the entire globe (excluding Antarctica) and more than 30% over large parts of North America and Europe under the high-emission scenario. This demonstrates the high sensitivity of SA to climate change, in which SA can exceed the natural range of climate variability and play a non-linear warming component role on the globe.

Since the onset of the Industrial Revolution, anthropogenic emission of greenhouse gases (GHG) has caused Earth's climate to warm measurably. Nonetheless, the rates of warming vary greatly across the globe[1–3]. Previous researches have revealed that regional variability is due in part to land–atmosphere coupling[4–7], whereby differences in land surface conditions (e.g., soil moisture, snowpack, and vegetation cover) modulate mass fluxes and energy to the atmosphere[8–16] and, furthermore, influence weather patterns and climate anomalies associated with GHG forcing. For instance, soil moisture–atmosphere coupling (SA) is linked to increased intensity and frequency of high-temperature extremes and heat waves via the impact of atmospheric warming on soil moisture in some regions[17–22]. However, a coherent picture of the extent to which global warming (including warming rate and heat extremes) will be impacted by SA and its time evolution under different emission scenarios remains unclear.

In order to analyze the SA effect on global warming, we employed six global climate models (CESM2, CMCC-ESM2, EC-Earth3, IPSL-CM6A-LR, MIROC6, and MPI-ESM1-2-LR) − the Land Surface, Snow and Soil Moisture Model Intercomparison Project (LS3MIP)[23], the Scenario Model Intercomparison Project (ScenarioMIP)[24], and the historical

experiment in Coupled Model Intercomparison Project phase 6 (CMIP6) − to isolate the climatic impact of SA during the extra-tropical summer under different warming scenarios. Experiments under the same emission scenario are driven by the same forcing agents (i.e., sea surface temperature, sea ice, and $CO_2$ concentrations). In the Land Feedback Model Intercomparison Project with prescribed Land Conditions experiment (LFMIP-pdLC) from LS3MIP, the soil moisture is fixed to its climatological state for the period 1980–2014 which is derived from historical global climate model output (Supplementary Fig. 1). Then the SA effect can be isolated from the relative differences between fully coupled experiments (historical, the SSP1-2.6, and SSP5-8.5 experiments) and LFMIP-pdLC experiment. In this study, we consider three time horizon periods: 1995–2014, 2040–2059, and 2080–2099 to represent modern, mid-term, and long-term future conditions, respectively.

## Results

Under each emission scenario investigated here, SA amplifies global warming over much of Earth's land surface (Fig. 1). Even under the most stringent GHG mitigation pathway, the SA-induced warming is

[1]Department of Atmospheric and Oceanic Sciences/Institute of Atmospheric Sciences, Fudan University, Shanghai, China. [2]Sino-French Institute for Earth System Science, College of Urban and Environmental Sciences, Peking University, Beijing, China. [3]Key laboratory of Cites' Mitigation and Adaptation to Climate Change in Shanghai, China Meteorological Administration, Shanghai, China. [4]State Key Laboratory of Severe Weather, Chinese Academy of Meteorological Sciences, Beijing, China. [5]These authors contributed equally: Liang Qiao, Zhiyan Zuo. ✉e-mail: zuozhy@fudan.edu.cn; rhzhang@fudan.edu.cn

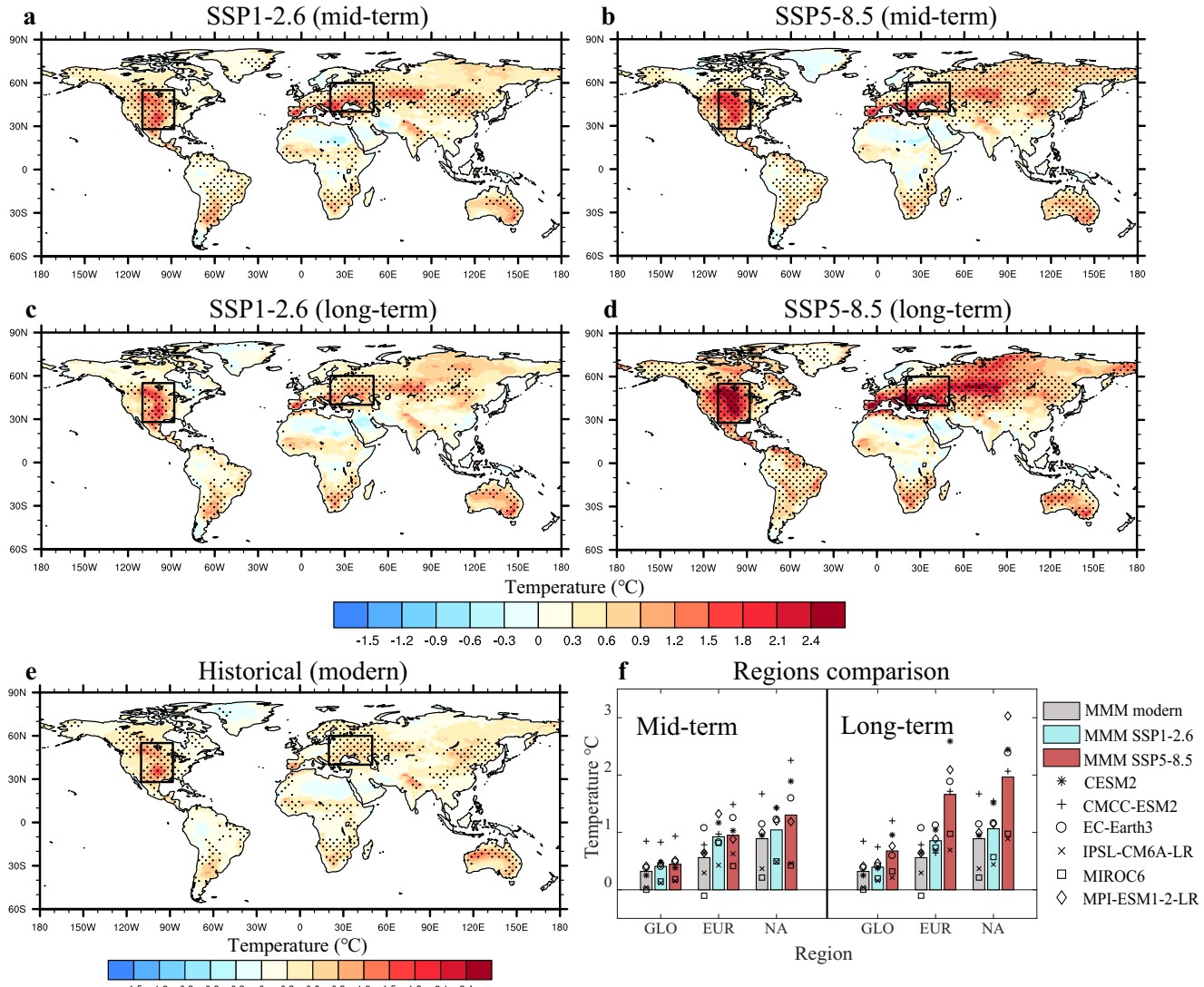

**Fig. 1 | Soil moisture-atmosphere coupling (SA) impact on surface air temperature (tas). a, b** represent spatial distributions of the SA influence on tas (°C, calculated the tas changes by subtracting fixed soil moisture experiment from fully coupled experiment under low-emission (SSP1-2.6) and high-emission (SSP5-8.5) scenarios for the mid-term future (2040–2059)). **c, d** represent same as for **a, b** but corresponding to the long-term future (2080–2099). **e** same as for **a**, but corresponding to the modern period (1995–2014). Black boxes in **a**–**e** represent central North America (NA: 28–55°N, 88–110°W) and central and eastern Europe (EUR: 40–60°N, 20–50°E). Black dots in **a**–**e** denote that the sign of the change is consistent with the sign of multi-model mean (MMM) in at least five of the six CMIP6 models. **f** is SA impact on global (GLO, excluding Antarctica), NA, and EUR tas under low- and high-emission scenarios for both mid-term and long-term future periods. Gray, blue, and red bars represent the MMM values for modern conditions, the low-emission (SSP1-2.6), and high-emission (SSP5-8.5) pathways, respectively. Individual models are represented by different types of points.

considerably greater than that experienced currently (Fig. 1a, c). Centers of SA-amplified warming occur primarily within the mid-latitudes of Northern Hemisphere and the subtropical regions of Southern Hemisphere, respectively. The most extreme warming occurs over central North America (NA: 28–55°N, 88–110°W) and central and eastern Europe (EUR: 40–60°N, 20–50°E) in modern period (0.89 ± 0.53 °C and 0.56 ± 0.41 °C), which is generally consistent with Berg et al[4]. The warming will be up to 2.4 °C in both regions in the long-term future under the very high-emission scenario (Fig. 1d). Under low-emission scenario, SA-induced warming varies little between the mid-term and long-term future projections; under the very high-emission pathway, however, warming is significantly more intense for long-term than for mid-term periods, particularly in EUR and NA. Specifically, global land-surface air temperature (excluding Antarctica) is projected to rise by 0.68 °C ± 0.38 °C owing to SA-induced warming by the end of this century, which is measurably higher than the mid-term projection (0.44 °C ± 0.28 °C; Fig. 1f). In NA and EUR, the difference between mid-

term and long-term future projections is 0.67 °C ± 0.66 °C and 0.71 °C ± 0.57 °C, respectively. The area over which SA-induced warming exceeds 0.5 °C (1.0 °C) would increase by nearly 30% (>200%) by the end of the 21st Century compared with mid-term projections (Supplementary Figs. 2–3).

Our findings exhibit a clear uptrend in SA-induced warming in the mid-latitude Northern Hemisphere under the very high-emission pathway, with comparatively weak trends under the lower-emission scenario (Fig. 2). For the very high-emission pathway, the positive trends caused by SA over EUR (0.17 °C ± 0.08 °C per decade) and NA (0.16 °C ± 0.09 °C per decade) explain 18.5 ± 9.7% and 18.8 ± 10.0%, respectively, of the overall warming rate in each region (Supplementary Table 1). Clearly, the uptrend of SA-induced warming associated with very high greenhouse gases emission would accelerate the speed of global warming, and the magnitude of acceleration would increase with time. Therefore, we posit that, unless we take early action to reduce emission, SA- and GHG-induced warming would become

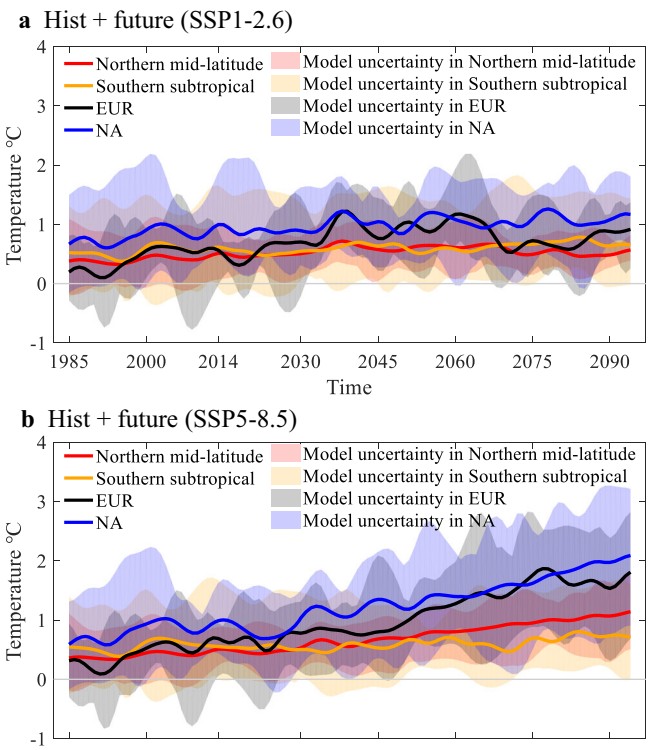

**a** Hist + future (SSP1-2.6)

**b** Hist + future (SSP5-8.5)

**Fig. 2 | Temporal evolution of soil moisture-atmosphere coupling (SA) driven surface air temperature (tas).** Temporal evolution of tas variability due to SA under (**a**) the low-emission scenario (SSP1-2.6) and (**b**) the high-emission scenario (SSP5-8.5). The time series is performed with Lanczos low-pass filtering to remove the interannual variability in 1980–2099. The total number of weights is 11 and the cut-off frequency of the low-pass filter is 1/11. For both periods, "Hist" denotes the period 1985–2014, and "future" is the period 2015–2094. The blue line is central North America (NA: 28–55°N, 88–110°W), the black line is central and eastern Europe (EUR: 40–60°N, 20–50°E), the red line is mid-latitudes of Northern Hemisphere (30–60°N, 180°W–180°E), and the orange line is subtropical regions of the Southern Hemisphere (20–40°S, 180°W–180°E). Shading represents the uncertainty among models.

closely coupled, resulting in a positive feedback that may hasten the approach of distinct climate range. Should the most stringent emission pathway be adopted, our results suggest SA-induced warming will weaken significantly (Fig. 2a).

Although SA contributes <10% to the likelihood of extreme high-temperature under modern conditions, this influence is likely to be strengthened over northern China and northernmost South America in the mid-term future (Fig. 3a). By the end of this century, the role of SA in the frequency and intensity of extreme high-temperature events globally is projected to rise significantly, consistent with the uptrend of SA-induced warming, and could approach 20% in the mid–high-latitude Northern Hemisphere and subtropical Southern Hemisphere (Fig. 3a). Over NA and EUR hotspots, SA would cause a rightward shift and pronounced flattening of the surface air temperature probability distribution function, representing an overall increase in the probability of extreme high-temperature over both regions (NA: +52.5%; EUR: +30.8%; Fig. 3c, d). Viewed another way, in the absence of SA-induced warming, the probability of the extreme high-temperature under the very high-emission pathway decreases by almost a third over NA and by a quarter over EUR by the end of the 21st Century.

In addition to frequency, the SA effect might also influence the intensity of extreme high-temperature; this process would impact North China and parts of South America in mid-term projections and globally in long-term projections (Fig. 3b). Our results indicate that the

intensity will rise by >1.5 °C globally and by as much as 8.0 °C over North America and Europe. Under the most stringent emission-mitigation scenario, the relationship between SA and the intensity and probability of extreme high-temperature would weaken (Supplementary Fig. 4), suggesting that reductions in GHG emission would actively reduce the severity of such events. We note that this feature is present in all the models considered here (Supplementary Figs. 5–6). The extreme high-temperature due to SA in India is diverse among different models though the SA-driven warming is significant, which may be associated with the large uncertainty of Indian summer monsoon precipitation[25].

GHG-driven warming is projected to dry the soil column[26] (Fig. 4a, b and Supplementary Fig. 7), thereby reducing evapotranspiration and allowing the ground surface to receive more solar shortwave radiation (longwave radiation is much weaker) through a reduction in cloud cover (Fig. 5a, b and Supplementary Fig. 8). Meanwhile, decreasing evapotranspiration could increase sensible heat flux from the land surface to the low-level atmosphere via decreasing the latent heat flux (Fig. 5c, d). The increasing sensible heat flux caused by the joint enhanced shortwave radiation and reduced latent heat flux trigger the nonlinear warming under severe GHG emission. Those phenomena are the strongest in the Northern mid-latitude, especially in EUR and NA, and Southern subtropical, where decreasing soil moisture can significantly change surface energy partitioning from latent heat flux to more sensible heat flux via decreasing evapotranspiration. On the contrary, there is a slight cooling effect caused by SA in a few regions, such as Sahara and Arabian Peninsula. The evapotranspiration change over those regions is negligible, while the negative cloud cover and positive shortwave radiation anomalies are generally significant, which suggests that the local evapotranspiration associated with soil moisture is not the primary factor dominating the surface energy balance. The impact of SA on surface energy over those regions may be due to the non-local effect and related to the large-scale circulation[16,27].

Under the very high-emission scenario, progressively drying soil column leads to an acceleration of the decline in evapotranspiration (Fig. 4b, d), with the result of increased positive radiative budgets and thereby the acceleration of the amplified-warming, particularly over NA and EUR. Moreover, the correlation coefficient between evapotranspiration and soil moisture will increase under a drying soil background, which eventually aggravates the reduction of evapotranspiration caused by soil drying over time (Fig. 4e–h). These characteristics are not only shown in the surface soil moisture which is most directly related to evapotranspiration, but also in the root zone soil moisture (Supplementary Fig. 9). The enhanced sensitivity of evapotranspiration to soil drying leads to the increase of SA-induced non-linear warming under very high GHG emission background. The non-linear increase of SA-induced warming, combined with the GHG-warming, will make global warming to act like a snowball, corresponding to SA and other climate factors approaching a novel and unpredictable statement.

Furthermore, we decompose the SA-induced changes in surface air temperature into the following radiative forcing terms: surface albedo, evapotranspiration, shortwave transmissivity, air emissivity, aerodynamic resistance, and residual terms[28] (see Methods). The primary driver of SA-induced warming is positive radiative forcing arising from the changes in evapotranspiration term and shortwave transmissivity term outlined above (Supplementary Figs. 10 and 11), which is the consequence of the decrease in evapotranspiration on shortwave radiation and sensible heat flux. Generally, the shortwave transmissivity term is larger than the evapotranspiration term for the global land in the modern period, while the effects of the two terms become equivalent in the future, especially in very high-emission scenario. Over EUR and NA in the long-term future, the combined positive radiation (sum of the above five terms) dominated by

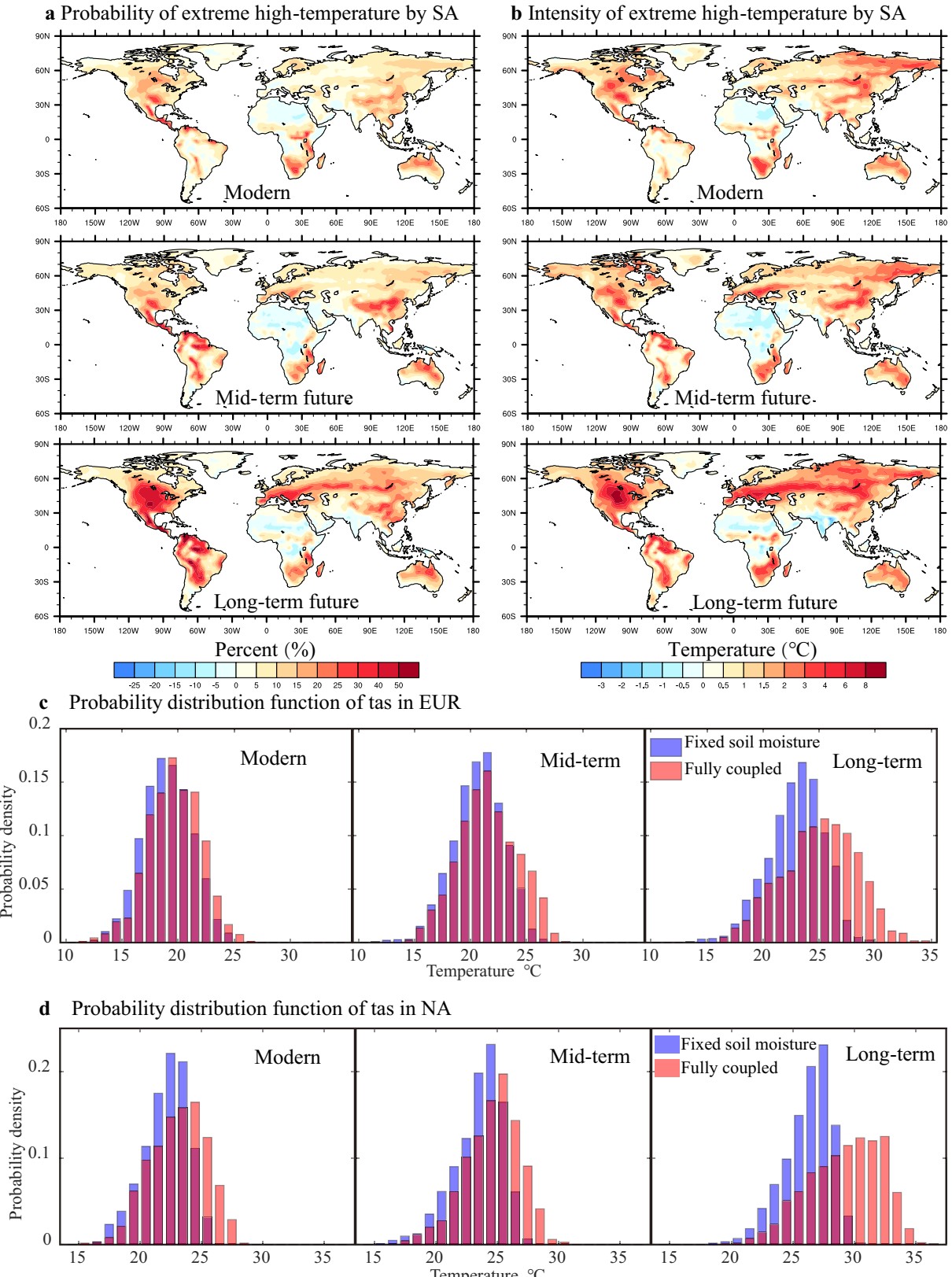

**Fig. 3 | Changes in the spatial distribution of 90th percentile of surface air temperature (tas) caused by soil moisture-atmosphere coupling (SA) and the probability distribution function of tas in the MPI-ESM1-2-LR model under the high-emission scenario. a** is spatial distribution of the change in probability of extreme high-temperature (%) caused by SA for the modern, mid-term, and long-term future periods (see Methods). **b** is same as **a** but for the intensity (°C) of extreme high-temperature events attributed to SA. **c**, **d** represent tas probability distribution functions under the high-emission scenario for central and eastern Europe (EUR: 40–60°N, 20–50°E) and central North America (NA: 28–55°N, 88–110°W) during the modern, mid-term, and long-term future periods.

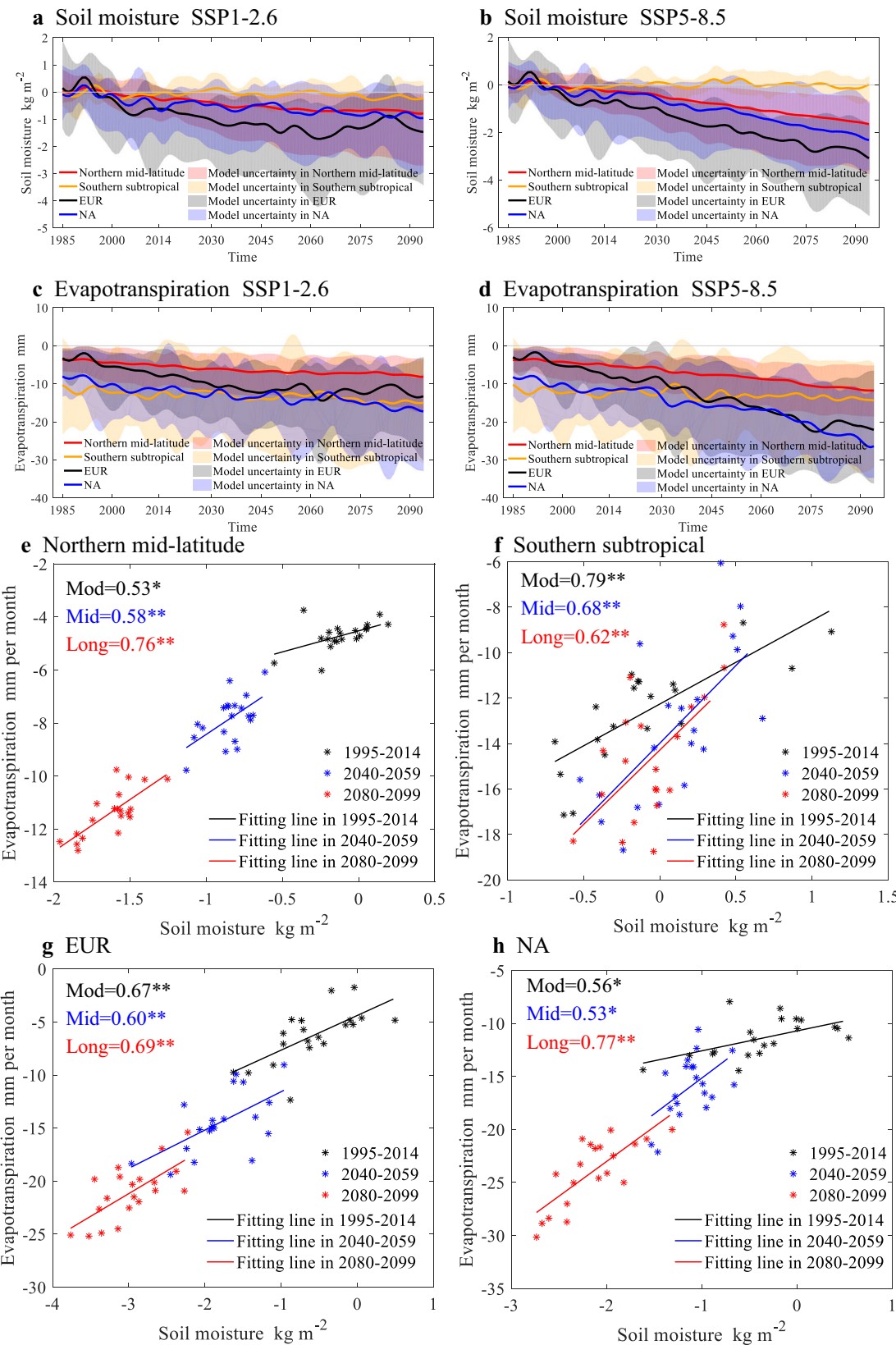

evapotranspiration term and shortwave transmissivity term will reach 33.5 ± 14.1 and 32.8 ± 16.7 W m⁻² under high-emission scenario. Compared with the modern period (15.4 ± 9.9 and 22.1 ± 9.2 W m⁻²), the combined radiation will increase by 117.5 ± 276.5% and 48.4 ± 53.3% over EUR and NA. However, under low-emission scenario, although the combined radiation will also increase (23.1 ± 5.9

and 25.3 ± 12.6 W m⁻²), they are smaller than half of those in high-emission scenario.

## Discussion

Output from CMIP6's LS3MIP and ScenarioMIP experiments projects that very high GHG emission will result in soil drying and reduced

**Fig. 4 | Soil moisture-atmosphere coupling (SA) mediated relationship between soil moisture and evapotranspiration. a, b** Represent temporal evolution of soil moisture (kg m$^{-2}$) due to SA under low- and high-emission scenarios. **c, d** same as **a, b**, but for evapotranspiration (mm per month). In **a–d**, the former (1985–2014) denotes modern conditions and the latter (2015–2094) denotes future conditions. The time series is performed with Lanczos low-pass filtering to remove the inter-annual variability in 1980–2099, the total number of weights is 11 and the cut-off frequency of the low-pass filter is 1/11. Shading is uncertainty among models. **e–h** Represent Scatter between soil moisture (kg m$^{-2}$) and evapotranspiration (mm per month) due to SA under high-emission scenario over northern middle latitudes: 30–60°N, 180°W–180°E, southern subtropical: 20–40°S, 180°W–180°E, central and eastern Europe (EUR): 40–60°N, 20–50°E, and central North America (NA): 28–55°N, 88–110°W. Numbers given in the upper left are correlation coefficients between evapotranspiration and soil moisture for the modern (Mod), mid-term (Mid), and long-term future (Long) periods. Two stars indicate that the correlation coefficient passes the 99% significance test, and one star indicates that the correlation coefficient passes the 95% significance test. Black, blue, and red lines are the fitting lines in the modern, mid-term, and long-term future periods.

evapotranspiration, thereby forcing more heat into the atmosphere via enhanced downward shortwave radiation and sensible heat flux. Such SA conditions will serve to further amplify the GHG-driven warming. Under the worst (highest) emission scenario, the amplification due to SA is projected to increase over time owing to the uptrend evapotranspiration rate associated with drying soil, which follows an accelerating amplified-warming. Such acceleration in SA-warming will make extreme high-temperature events both more frequent and more severe, particularly over North America and Europe. The implication of these findings suggests that mitigation efforts corresponding to acceleration of SA-driven warming must be implemented at an early stage to minimize the risk of climate shock. Policymakers must also consider maintaining ecosystem stability as a tool for sustaining soil moisture within appropriate limits, thereby avoiding the worst impacts of elevated SA, especially in North America and Europe. Here we emphasize on the local effect of SA on surface air temperature while the latest research shows that SA can also affect the large-scale atmospheric circulation under the global warming background[16]. In this view, the joint contribution of local and non-local SA effect on global land warming needs to be further investigated. Finally, the results may have some projection uncertainty considering the limitations of the parameterization in SA[29–31] and scenario uncertainty due to the lack of knowledge of future radiative forcing[32]. Nevertheless, the models we used in this research have high reliability in the historical simulation of soil moisture[33], and we reduced the uncertainty by multi-model mean. Meanwhile, different single models have similar results for the acceleration in SA-driven warming in the future projection. So, the conclusion that SA amplifies greenhouse gas-driven global warming is relatively reliable and robust.

## Methods
### CMIP6 models
Six global climate models in CMIP6 (CESM2, CMCC-ESM2, EC-Earth3, IPSL-CM6A-LR, MIROC6, and MPI-ESM1-2-LR) are used to analyze the impact of SA on surface air temperature. Extra-tropical summer is defined as the June–August (JJA) average in Northern Hemisphere and the December–February (DJF) average in Southern Hemisphere. Since there have been only six CMIP6 LS3MIP models published so far, we only use these six models in this article. The multi-model mean is used to remove uncertainties arising from model differences. For the majority of our analyses, we incorporated monthly data derived from all six models. Due to the lack of latent heat flux in MIROC6 model, the multi-mode mean of latent heat flux is the result of the other five models. The surface air temperature probability distribution function and extreme high-temperature are calculated from daily data for each model (no multi-model mean) separately. Because the amount of monthly data is not enough compared with daily data, we are concerned that the probability distribution function characteristics cannot be accurately captured. Meanwhile, the multi-model mean will mask the intensity distribution of extreme high-temperatures when the number of models is little. So, we used daily data to calculate the probability distribution function and extreme high-temperature probability for each model separately in Fig. 3 (MPI-ESM1-2-LR model under high-emission scenario), Supplementary Fig. 4 (MPI-ESM1-2-LR model under

low-emission scenario), Supplementary Fig. 5 (CMCC-ESM2), and Supplementary Fig. 6 (IPSL-CM6A-LR). And the results showed that they had similar characteristics. In the six global climate models, CESM2, CMCC-ESM2, IPSL-CM6A-LR, and MPI-ESM1-2-LR include simulated dynamic vegetation, and EC-Earth3 and MIROC6 models do not include simulated dynamic vegetation. The surface soil moisture depth is 10 cm, and the root zone soil moisture depth is 100 cm.

The LFMIP-pdLC experiment in LS3MIP, SSP1-2.6,and SSP5-8.5 experiments in ScenarioMIP, and the historical experiment are analyzed. The LFMIP-pdLC experiment in LS3MIP is used to assess the impact of land-atmosphere coupling caused by soil moisture on weather and climate through fixing the soil moisture as their climatological state (the annual mean cycle for the period 1980–2014 derived from historical global climate model output). Then the SA effect can be isolated through the relative differences between fully coupled experiments (historical, the SSP1-2.6, and SSP5-8.5 experiments) and fixed soil moisture experiment (LFMIP-pdLC experiment)[23]. The ScenarioMIP reflects future climate (2015–2099) under high and low Shared Socioeconomic Pathways (SSP)[24]. The SSP1-2.6 experiment and *r1i1p1f1* member in the LFMIP-pdLC experiment corresponds to the same low-emission scenarios (2015–2099), and the SSP5-8.5 experiment and *r1i1p1f2* member in the LFMIP-pdLC experiment corresponds to the same high-emission scenarios (2015–2099).

### Extreme high-temperature
The spatial distribution of extreme high-temperature (Fig. 3a, b, Supplementary Fig. 4a, b, Supplementary Fig. 5a, b, and Supplementary Fig. 6a, b) is determined by calculating the 90th percentile of surface air temperature in each grid point. And for the probability distribution function of surface air temperature over NA and EUR (Fig. 3c, d, Supplementary Fig. 4c, d, Supplementary Fig. 5c, d, and Supplementary Fig. 6c, d), regional average of surface air temperature is calculated first, and then the probability distribution function of the regional average is calculated. In order to analyze the extreme high-temperature changes driven by SA, the 90th percentile of surface air temperature in LFMIP-pdLC experiment is used as the threshold temperature in three time periods (modern, mid-term, and long-term) separately. The probability and intensity differences of surface air temperature in coupled experiment (historical, the SSP1-2.6, and SSP5-8.5 experiments) relative to the threshold temperature in LFMIP-pdLC experiment are taken as the SA effect across the three time horizon periods (modern, mid-term, and long-term).

### Contribution to the warming trend
We calculated the warming trend (°C per decade, 2015-2099) of total, GHG, and SA effect on ground surface air temperature under high-emission scenario (SSP5-8.5) over the globe (excluding Antarctica), northern middle latitudes (Northern: 30–60°N, 180°W–180°E), southern subtropical latitudes (Southern: 20–40°S, 180°W–180°E), Europe (EUR: 40–60°N, 20–50°E), and North America (NA: 28–55°N, 88–110°W) in future projections (Supplementary Table 1). (1) The total contribution to the warming trend is obtained by calculating the trend of the surface air temperature in fully coupled experiment under high-emission scenario (SSP5-8.5), because the time series of this

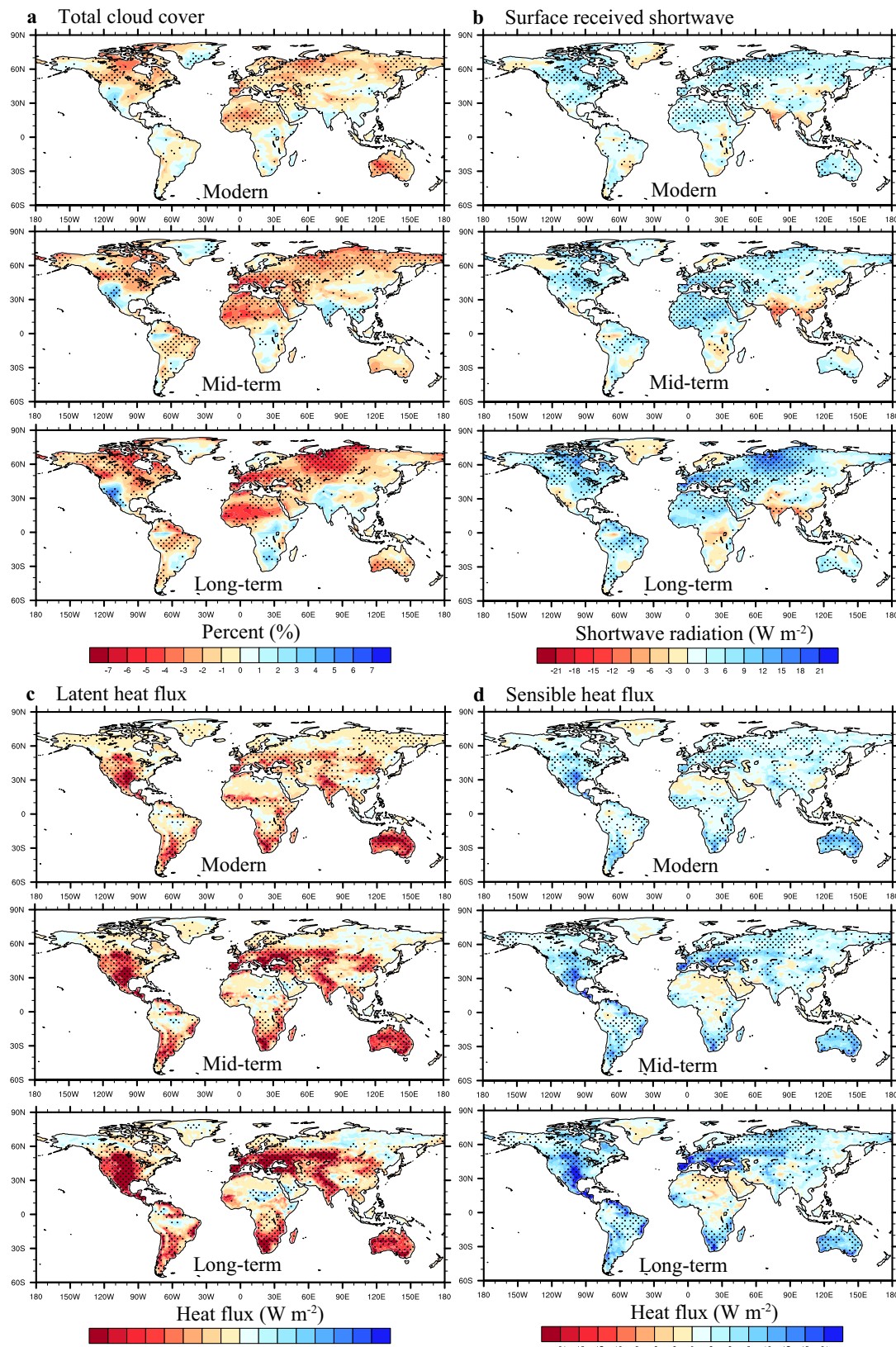

**Fig. 5 | Spatial distributions of land surface meteorological elements caused by soil moisture-atmosphere coupling (SA) in modern (1995–2014), mid-term future (2040–2059), and long-term future (2080–2099) periods under the very high-emission scenario (SSP5-8.5). a** is total cloud cover (%), **b** is surface-received shortwave radiation (W m⁻²), **c** is latent heat flux (W m⁻²), and **d** is sensible heat flux (W m⁻²). Black dots signify agreement between the sign of change and the multi-model mean in at least five of the six (or four of five) CMIP6 models.

experiment included both GHG and SA effects; (2) the contribution of SA to the warming trend is obtained by calculating the trend of the difference of surface air temperature between fully coupled experiment (SSP5-8.5) and fixed soil moisture experiment (LFMIP-pdLC) under high-emission scenario, because the time series of this difference between the two experiments considered as SA effect; (3) the contribution of GHG to the warming trend is obtained by calculating the trend of the surface air temperature in fixed soil moisture experiment under high-emission scenario (LFMIP-pdLC), because the time series of this experiment included GHG effect and excluded SA effects.

### Decomposition of SA-driven changes in surface air temperature[28]

We use the surface energy balance to decompose the surface air temperature ($T_a$) changes caused by SA. The changes in $T_a$ are mainly driven by the land surface temperature ($T_s$) and atmospheric circulation ($\Delta T_a^{cir}$), where $T_s$ interacts with $T_a$ through radiative and non-radiative fluxes ($\Delta T_a^{rad}$). So, the $\Delta T_a$ can be expressed as:

$$\Delta T_a = \Delta T_a^{rad} + \Delta T_a^{cir} \qquad (1)$$

In order to calculate the change of $T_s$ caused by SA, we start with the formula of land surface energy balance:

$$S_n + L_n = \lambda E + H + G \qquad (2)$$

where $S_n$ and $L_n$ are the net short-wave and long-wave radiation at the surface. $\lambda$, $E$, and $H$ are the vaporization latent heat, evapotranspiration, and sensible heat flux. $G$ is the ground heat flux, which magnitude is relatively small and can be ignored in seasonal and longer timescales. Equation (2) can be simplified as:

$$S_n + L_n = \lambda E + H \qquad (3)$$

$S_n$ can be expressed as:

$$S_n = S\tau(1 - \alpha) \qquad (4)$$

where $S$ is the solar radiation flux at atmosphere top, $\tau$ is the atmospheric short-wave transmissivity, $\alpha$ is the surface albedo.

According to the Stephan–Boltzmann law, the downward long-wave radiation at the land surface can be calculated roughly as:

$$L_\downarrow = \varepsilon_a \sigma T_a^4 \qquad (5)$$

where $\varepsilon_a$ is atmospheric air emissivity, $\sigma$ is the Stephan–Boltzmann constant ($5.67 \times 10^{-8}$ Wm$^{-2}$ K$^{-4}$).

The upward long-wave radiation is expressed as:

$$L_\uparrow = (1 - \varepsilon_S)\varepsilon_a \sigma T_a^4 + \varepsilon_S \sigma T_S^4 \qquad (6)$$

where $\varepsilon_S$ is the land-surface emissivity, and it can be treated as a constant of 0.95, since the $\varepsilon_S$ varies very little over different land covers, around 0.95. Therefore, the net long-wave radiation at the land surface is expressed as:

$$L_n = L_\downarrow - L_\uparrow = \varepsilon_s \sigma \left( \varepsilon_a T_a^4 - T_s^4 \right) \qquad (7)$$

The sensible heat flux is expressed as:

$$H = \frac{\rho C_d (T_s - T_a)}{r_a} \qquad (8)$$

where $\rho$ is the air density (1.21 kg m$^{-3}$), $C_d$ is the air specific heat at constant pressure (1013 J kg$^{-1}$ K$^{-1}$), and $r_a$ is the aerodynamic resistance. Because in water-restricted areas such as arid and semi-arid regions, the change of latent heat flux is more determined by soil moisture, so latent heat flux is not written as a function of the change of $T_a$.

Using Eqs. (2)–(8), the surface energy balance equation can be expanded as following:

$$S\tau(1 - \alpha) + \varepsilon_s \sigma \left( \varepsilon_a T_a^4 - T_s^4 \right) = \lambda E + \rho C_d \frac{(T_s - T_a)}{r_a} \qquad (9)$$

Assuming that S, $\lambda$, $\rho$, $C_d$, $\sigma$ and $\varepsilon_s$ are independent of $T_s$, it can be further differentiated the Eq. (9) with respect to $T_s$:

$$\Delta T_s = \frac{1}{f_s} \left( -S\tau\Delta\alpha - \lambda\Delta E + S(1-\alpha)\Delta\tau + \varepsilon_s \sigma T_a^4 \Delta\varepsilon_a + \frac{\rho C_d (T_s - T_a)}{r_a^2} \Delta r_a \right) + \frac{\rho C_d/r_a + 4\varepsilon_S \sigma \varepsilon_a T_a^3}{\rho C_d/r_a + 4\varepsilon_S \sigma T_s^3} \Delta T_a \qquad (10)$$

where $f_s$ is an energy redistribution factor:

$$f_s = \rho C_d/r_a + 4\varepsilon_s \sigma T_s^3 \qquad (11)$$

$f_s^{-1}$ is the $T_s$ sensitivity to 1 W m$^{-2}$ radiative forcing.

On the right side of Eq. (10), the first term is the $T_s$ change caused by radiative and thermodynamic forcings associated with SA-driven changes in surface albedo, evapotranspiration, short-wave transmissivity, air emissivity, and aerodynamic resistance ($\Delta T_s^{rad} = (1/f)(-S\tau\Delta\alpha - \lambda\Delta E + S(1-\alpha)\Delta\tau + \varepsilon_s \sigma T_a^4 \Delta\varepsilon_a + \frac{\rho C_d(T_s - T_a)}{r_a^2}\Delta r_a)$). The second term ($\frac{\rho C_d/r_a + 4\varepsilon_S \sigma \varepsilon_a T_a^3}{f_s}\Delta T_a$) quantifies the coupling strength between $T_a$ and $T_s$. It means that $T_s$ varies with $T_a$ caused by SA-driven change in air advection ($\Delta T_s^{cir} = \frac{\rho C_d/r_a + 4\varepsilon_S \sigma \varepsilon_a T_a^3}{f_s}\Delta T_a^{cir}$). Meanwhile, it also suggests that the $T_s$ change further drives a change in $T_a$ through the surface heating rate change ($\Delta T_a^{rad} = \frac{f_s}{\rho C_d/r_a + 4\varepsilon_S \sigma \varepsilon_a T_a^3}\Delta T_s^{rad}$). Through equations (1) and (10), $T_a$ changes caused by SA can be decomposed into the following forms:

$$\Delta T_a = \frac{1}{f} \left( -S\tau\Delta\alpha - \lambda\Delta E + S(1-\alpha)\Delta\tau + \varepsilon_s \sigma T_a^4 \Delta\varepsilon_a + \frac{\rho C_d (T_s - T_a)}{r_a^2}\Delta r_a \right) + \Delta T_a^{cir} \qquad (12)$$

Where $f = \rho C_d/r_a + 4\varepsilon_s \sigma \varepsilon_a T_a^3$, $f^{-1}$ is the $T_a$ sensitivity to 1 W m$^{-2}$ radiative forcing.

The change in surface air temperature ($\Delta T_a$) caused by SA is decomposed into radiative forcing terms by Eq. (12)[28] and incorporating surface albedo ($-S\tau\Delta\alpha$), evapotranspiration ($-\lambda\Delta E$), shortwave transmissivity ($S(1-\alpha)\Delta\tau$), air emissivity ($\varepsilon_s \sigma T_a^4 \Delta\varepsilon_a$), aerodynamic resistance ($\frac{\rho C_d(T_s - T_a)}{r_a^2}\Delta r_a$), and residual term ($\Delta T_a^{cir}$). In the decomposition of surface air temperature, we employed four modes of multi-model mean (CESM2, CMCC-ESM2, EC-Earth3, and IPSL-CM6A-LR) in our calculations. The remaining two models (MIROC6 and MPI-ESM1-2-LR) were omitted because they lack the relevant radiation input data.

## Data availability
All data used in this study are freely available online. The CMIP6 model simulations are from https://esgf-node.llnl.gov/search/cmip6/.

## Code availability
Analysis and figure generation were performed using NCL and MATLAB. The code and scripts of the five figures in the paper are available from Zenodo: https://doi.org/10.5281/zenodo.7928584.

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

## Acknowledgements

We acknowledge the World Climate Research Programme's Working Group on Coupled Modelling, which is responsible for CMIP, and thank the various climate modeling groups who produced and made available their model output. Z.Z. acknowledges the National Key Research and Development Program (Grant No. 2022YFF0801703) and the National Natural Science Foundation of China (41822503). D.X. acknowledges the National Natural Science Foundation of China (42175053).

## Author contributions

Z.Z., L.Q., R.Z., and S.P. conceived of and designed the study, L.Q. performed analyses, Z.Z. and L.Q. wrote the paper. D.X. and K. Z.

assisted in the framing and development of ideas. Z.Z., L.Q., R.Z., S.P., D.X., and K.Z. edited the paper.

## Competing interests
The authors declare no competing interests.
