## [Peer Review File · Nature Communications]

Soil moisture–atmosphere coupling accelerates global warmingREVIEWER COMMENTS

Reviewer #1 (Remarks to the Author):

This study presented potentially original findings on the role of land-atmosphere coupling as an amplifier of GHG warming over some regions, with this amplifying effect stronger under the high emission scenario. The presentation is overall clear.

However, before the manuscript can be accepted, the authors need to address the following major concerns (about the interpretation of results and the methodology used) and minor comments.

Major concerns:

1) While the authors emphasized the positive LA contributions to warming over NA and EUR, they didn't explain why these positive LA contributions are strong primarily over these regions. In fact, there are also negative LA contributions to warming over Africa (except southern part) in Fig. 1. The authors need to analyze and explain the results over Africa (except southern part) in a similar way (to ensure that the authors' arguments are robust).

Similarly, the LA contributions to extreme high temperatures are negative over most of Africa and India (Fig. 3). Some explanations are also needed.

2) Lines 356-365 discussed the method to compute GHG and LA contributions to the total warming, but it is not obvious what these terms exactly represent. The authors need to present and discuss these individual contributions for the "modern" period. From Fig. 2, it seems that the LA-driven temperature difference (warming) is quite large over NA in the "modern" period. The question is: is this LA-driven large difference realistic? – at least some comparisons with any prior studies are needed.

To further ensure that the methodology is appropriate, the authors should add the results for the period of 1900-1919 (when GHG forcing is weak) - if such model simulation results are available.

3) Eq. (1) is the primary equation used to explain the mechanisms for the results, and hence the assumptions used to derive this equation should be presented. For instance, it is unclear how the downward and upward longwave radiation variations are computed.

Minor comments:

4) the use of the term "Snowball effect" in the title does not seem appropriate, as there are both positive and negative LA contributions to warming (see major comment 1).

5) for Table S1, provide the scenario used. Furthermore, provide the range (i.e., min and max) for each value (from the various model results).

6) mention in the text how the "globe" is defined? – does it include Antarctic and Greenland?

7) Lines 333-337: provide more details on LFMIP-pdLC.

Reviewer #2 (Remarks to the Author):

The paper "Snowball effect of land-atmosphere coupling on global warming" explores the role of land-atmosphere feedbacks on projections of surface air temperature by comparing the outputs from LS3MIP experiment with the historical and future projections with SSP1-2.6 and SSP5-8.5. The results indicate that the land surface has a compounding effect on the global warming due to the feedbacks between the land and the atmosphere. Specifically, over high latitudes, the fully coupled models show a decrease in evaporation and decrease in cloud cover which cause an increase in radiation at the surface and an increase in sensible heating that led to an increase in

surface air temperature and compounds the effects of greenhouse gasses.

The paper is logically organized which makes it easy to follow. The experiment design is interesting, and the results provide insights into the coupling between the land and the atmosphere and its impact on future projects of temperature. Based on this, the paper is well suited for Nature Communications and merits publication. Despite these positive aspects, the paper is not particularly clear in defining certain aspects of the work and there needs to be some discussion about the uncertainty in the models. Based on this, here are a few suggestions for improvement.

First, there are a few places in the paper that need some clarification. For example, in lines 50-56 it provides a brief overview of the LS3MIP experiment and how the models are used to derive the results. I found this section to be confusing and it would benefit for a little bit more detail about the LS3MIP experiment. Specifically, it uses the climatology from 1980-2014 for soil moisture, but does it also change soil temperature, snow cover and other land surface states? What impacts does changing some aspects of the land surface model but not others have on the model simulations?

Line 104-106 is also a bit confusing. What is meant by "LA will switch from its role as an agent of internal climate variability to one in which LA is tied directly to GHG emissions"? This statement is problematic on many levels and should be revised. It is unclear on what is meant by "agent" and how LA is "tied directly to GHG".

Line 329 was confusing as it is unclear by what is meant by "monthly data will affect the distribution of extreme temperature probability". Is this referencing the fact that the monthly data will not show as high of a value as daily data due to temporal averaging? If so, then it is still unclear exactly how this was resolved given that some models provide daily data and others only have monthly data.

Lines 345-351 are very confusing, and I am not completely sure I understand what is going on. Are you doing a regional scale analysis (e.g., multiple grid cells within a certain range of the center grid cell) or is it just a grid cell analysis? Using without LA interactions as the threshold temperature only works if you are comparing the differences between without LA and with LA across the three time periods (modern, mid-term and long-term). I think that is what is being done here, but it is difficult to understand.

Extended Data Fig. 1 (Lines 412-415) was not helpful and difficult to see any relationship given the monthly frequency of the data. Consider revising so that it can clearly show the difference between the two model runs (uncoupled, coupled) that you are trying to emphasize here.

Extended Data Fig. 9 (Lines 471-481) was very interesting and clearly shows the attribution of the change from the coupled and uncoupled models. Consider trying to highlight this more in the paper.

Lastly, and most importantly, there needs to be a broader discussion about the uncertainty of LA coupling in models and how it has the potential to spin out of control and not provide realistic representation of coupling (Dirmeyer, 2013; Roundy et al., 2013; Santanello et al., 2015). Therefore, there is always the potential that what is being shown here is not necessarily an indication of what will happen in a future climate with increased GHG, but could just be coupled models snowballing into an unrealistic coupling state due to limitations of parameterization in the land surface and atmospheric models.

Dirmeyer, P. a. (2013). Characteristics of the water cycle and land-atmosphere interactions from a comprehensive reforecast and reanalysis data set: CFSv2. *Climate Dynamics*, 41(3-4), 1083-1097. <https://doi.org/10.1007/s00382-013-1866-x>
Roundy, J. K., Ferguson, Craig R., & Wood, Eric F. (2013). Impact of land-atmospheric coupling in

CFSv2 on drought prediction. *Climate Dynamics*, 1–14. <https://doi.org/10.1007/s00382-013-1982-7>

Santanello, J. A., Roundy, J., & Dirmeyer, P. A. (2015). Quantifying the Land–Atmosphere Coupling Behavior in Modern Reanalysis Products over the U.S. Southern Great Plains. *Journal of Climate*, 28(14), 5813–5829. <https://doi.org/10.1175/JCLI-D-14-00680.1>

Reviewer #3 (Remarks to the Author):

Review for „Snowball effect of land–atmosphere coupling on global warming“ by Qiao et al.

Summary

=====

Qiao et al. analyse 6 CMIP6 climate model simulations under 2 emission scenarios and the historical period in comparison with their LFMIP-pdLC simulations under LS3MIP to isolate and study the impact of LA coupling on the warming in the 21st century from the overall warming. The authors show that the impact of soil moisture and snow cover change due to GHG changes will enhance the global warming in case of the high emission scenario.

The topic of the study is relevant for understanding the contribution of LA coupling to climate change in global climate model simulations under different GHG scenarios. But it needs major revision prior to publication.

Major comments:

=====

1) The study complements the publication Zhou, S., Williams, A.P., Lintner, B.R. et al. Diminishing seasonality of subtropical water availability in a warmer world dominated by soil moisture–atmosphere feedbacks. *Nat Commun* 13, 5756 (2022). <https://doi.org/10.1038/s41467-022-33473-9>, however this study is not cited.

2) The title is gimmicky („snowball effect“) and not covering the study („effect of land–atmosphere coupling“), the study only analyses the soil moisture and snow cover effect on the global warming, not the land-atmosphere coupling effect.

3) This study needs to account at least for the root zone soil moisture. Showing the 10 cm soil moisture in Fig 4 is not sufficient for the goal of this study, it needs to be at least 1m. E.g. Fig. 4 shows only about 1-2 kg/m² in soil moisture change 100 years, i.e. 1-2 mm in 10 cm of soil, i.e. 2.5-5% of the porosity. Further Evapotranspiration uses the root zone, not the top 10 cm.

4) It is necessary to distinguish regions with latent heat flux limited by soil moisture and those limited by radiation, because they will result in different land atmosphere coupling paths. Only under soil moisture limitation the soil moisture impacts the temperature. This is very well seen in the results by Qiao et al. but not discussed. Include this in the analyses.

In Detail:

=====

1. „Snowball“ in the frame of climate research is associated with the „Snowball Earth“ and therefore the „snowball effect“ even though meaning something different may lead to the wrong attention. Please change the title to fit the study (see summary) and not to be gimmicky.

2. Lines 44-48: Something is missing here. It is not clear how many models etc without jumping to the methods section

3. Line 51: LFMIP simulations are not „uncoupled“. It is important to name here LFMIP so nobody is confused with uncoupled LMIP (Bart Van den Hurk et al. *Geoscientific Model Development*, DOI 10.5194/gmd-9-2809-2016)

4. Change lines 55-56, e.g.: „For the LFMIP experiment, land-surface forcing comprises the annual mean cycle of soil moisture and snow cover for the period 1980–2014 derived from historical global climate model output by fixing their climatological state to remove long-term trends and inter-annual variability (Extended Data Fig. 1)“. This is because not the land surface state but the soil moisture and snow cover are fixed, which is essential for the analyses because this means that the surface energy fluxes and land surface temperature can change in LFMIP.

5. Lines 55: the LA effect cannot be isolated, only that of soil moisture and snow cover.

6. Line 57-58: Why do you use 20-year climatologic periods instead of 30-year periods? You are looking at soil moisture and ET changes, not only temperature. See e.g. Liersch et al 2020 *Environ. Res. Lett.* 15 104014 DOI 10.1088/1748-9326/aba3d7 and Hawkins, E., & Sutton, R. (2016). Connecting Climate Model Projections of Global Temperature Change with the Real World, *Bulletin of the American Meteorological Society*, 97(6), 963-980. Retrieved Feb 28, 2023, from <https://journals.ametsoc.org/view/journals/bams/97/6/bams-d-14-00154.1.xml>

7. Line 60/61: Change to „Under each emission scenario investigated here, LA due to soil moisture and snow cover amplifies global warming over much of Earth’s land surface (Fig. 1).“

8. Lines 63-68: Mention here the reasons for those regions. It is not surprising to find the amplification in regions which will become increasingly soil moisture limited in evapotranspiration.

9. SAT: Use the ESGF abbreviation tau_s, not SAT, which is rather uncommon.

10. Line 337: Delete „Such as“

11. All Figs with „LA induced“ differences (e.g. Fig. 4, Extended Figs. 7,8,10): Name that these are CMIP6 and LFMIP difference plots (as in Fig. 1)

12. Line 34: Too many references, some are doubling the statement. References 2,3,6 and 7 are superficial.

13. Reference 8 has the wrong first author. It should be Berg et al.

14. Line 41: Decide to cite reference 22 or 24 because they are not providing any extra information needed for your study.

15. Figure 8: Why is figure 8 only in the extended data when your title is about land-atmosphere coupling?

16. How is the albedo changing? Due to snow and moisture or also due to dynamic vegetation in the models?

17. Do the models include simulated dynamic vegetation (i.e. LAI growth and root growth due to atmospheric and soil moisture conditions)? Add this information.

Responses to NCOMMS-22-53760-A

Reviewer #1:

Reviewer Comment

This study presented potentially original findings on the role of land-atmosphere coupling as an amplifier of GHG warming over some regions, with this amplifying effect stronger under the high emission scenario. The presentation is overall clear.

However, before the manuscript can be accepted, the authors need to address the following major concerns (about the interpretation of results and the methodology used) and minor comments.

Reply

We thank the reviewer for the positive comments on original findings of this manuscript. We have revised the manuscript in accordance with the reviewer's comments. Below please find our point-to-point response to the comments.

Major Comments:

Reviewer Comment

1. While the authors emphasized the positive LA contributions to warming over NA and EUR, they didn't explain why these positive LA contributions are strong primarily over these regions. In fact, there are also negative LA contributions to warming over Africa (except southern part) in Fig. 1. The authors need to analyze and explain the results over Africa (except southern part) in a similar way (to ensure that the authors' arguments are robust). Similarly, the LA contributions to extreme high temperatures are negative over most of Africa and India (Fig. 3). Some explanations are also needed.

Reply

Firstly, we changed title to "Soil moisture-atmosphere coupling accelerates global warming" in the revised manuscript because the LFMIP-pdLC experiment only reflects the effect of fixed soil moisture.

The warming effect of soil moisture-atmosphere coupling (SA) is most intensive over NA and EUR, so we emphasize the SA-warming in the two regions. About the

relevant explanation, we have illustrated by Figs. 4-5 and Supplementary Figs. 8-11. Actually, this manuscript is organized as two parts: Part 1 is about facts (Figs. 1-3; Supplementary Figs. 1-7) and Part 2 is about the relevant explanations (Figs. 4-5; Supplementary Figs. 8-11).

Fig. 4 demonstrates the most weakening of evapotranspiration in NA and EUR, which conduce to the most increase of sensible heat flux because of the intensive SA over the two regions. If we consider the net energy of ground surface and low-atmosphere as a cake, the less evapotranspiration increases surface receiving shortwave radiation (the anomalous longwave radiation is much weaker, Supplementary Fig. 8 in the revised manuscript) through a reduction in cloud cover (Figs. 5a-b in the revised manuscript), which enlarges the energy cake. On the other way, the less evapotranspiration also leads to more energy transferred from latent heat flux to the sensible heat flux (Figs. 5c-d in the revised manuscript), especially over those dry-wet transition regions such as NA and EUR. Bigger slice of the bigger cake conduces to the strongest positive sensible heat flux over NA and EUR, which thereby become a soil moisture-warming hotspot. Following the comment, we have strengthened the explanation in the revised manuscript (Page 11 Lines 166-177: GHG-driven warming is projected to dry the soil column²⁶ (Figs. 4a-b and Supplementary Fig. 7), thereby reducing evapotranspiration, allowing the ground surface to receive more solar shortwave radiation (longwave radiation is much weaker) through a reduction in cloud cover (Figs. 5a-b and Supplementary Fig. 8). Meanwhile, decreasing evapotranspiration could increase sensible heat flux from the land surface to the low-level atmosphere via decreasing the latent heat flux (Figs. 5c-d). The increasing sensible heat flux caused by the joint enhanced shortwave radiation and reduced latent heat flux conduce to the nonlinear warming under severe GHG emission. Those phenomena are the strongest in the Northern mid-latitude and Southern subtropical, especially in EUR and NA, where decreasing soil moisture can significantly change more surface energy allocation from latent heat flux to sensible heat flux via decreasing evapotranspiration.).

Fig. 5| Spatial distributions of land surface meteorological elements caused by SA in modern (1995-2014), mid-term (2040-2059), and long-term future (2080-2099) periods under the very high-emission scenario (SSP5-8.5). a is total cloud cover (%), b is surface-received shortwave radiation (W/m^2), c is latent heat flux (W/m^2), and d is sensible heat flux (W/m^2). Black dots signify agreement between the sign of change and the multi-model mean in at least five of the six (or four of five) CMIP6 models.

Supplementary Fig. 8| Same as Fig. 5, but for evapotranspiration (mm) and surface received longwave (W/m^2)

Secondly, the negative warming effect of SA over African Sahara is much weaker in comparison with those positive warming, we did not specially discuss the phenomenon in the manuscript. The soil moisture anomalies over Sahara is generally very small due to the extremely dry condition and the models may not capture the plausibly local impact of SA. On the other way, based on the Fig. 5 and Supplementary Fig. 8 in the revised manuscript, we can find that the evapotranspiration anomalies over Sahara is very little while the negative cloud cover and positive shortwave radiation anomalies are generally significant, which suggests that the local evapotranspiration associated with soil moisture is not the major factor dominating the surface energy. The impact of SA on the surface energy over Sahara is non-local and related to the large-scale circulation anomalies, which may be the reason for the positive shortwave radiation whereas negative sensible heat flux anomalies with the local little evapotranspiration anomaly (Berg et al., 2017; Zhou et al., 2022). Overall, the negative sensible heat flux, which may due to the non-local LA impact, conduces to the negative surface air temperature anomalies and less extreme high-temperature events over Sahara. Following the suggestion, we add the relevant content in the revised manuscript (Pages 11-12 Lines 177-183: On the contrary, there is a slight cooling effect caused by

SA in a few regions, such as Sahara and Arabian Peninsula. The evapotranspiration change over those regions is very little, while the negative cloud cover and positive shortwave radiation are generally significant, which suggests that the local evapotranspiration associated with soil moisture is not the primary factor dominating the surface energy. The impact of SA on surface energy over those regions may be due to the non-local effect and related to the large-scale circulation (Berg et al., 2017; Zhou et al.,2022).)

Finally, about the negative extreme high-temperature anomalies over India in long-term future, as pointed out by the reviewer, we do not expect it because the surface air temperature anomaly is positive though it is not significant in the multi-model mean (Figs. 1c-d). Furthermore, there are great differences among different models in India (negative in MPI-ESM1-2-LR (Figs. 3a-b), positive in CMCC-ESM2 (Supplementary Figs. 5a-b) and little change in IPSL-CM6A-LR (Supplementary Figs. 6a-b)), which may be the consequence of the large uncertainty of Indian summer monsoon precipitation (Prasanna 2016).

In MPI-ESM1-2-LR model, the SA negative effect on extreme high-temperature over India and North Africa can be explain through the morphological changes of PDF (Explained Fig. 1). Specifically, the PDF of surface air temperature over India and North Africa does not change significantly by SA, and shifted slightly to the left, especially in the long-term future, which suggests that the SA effect on extreme high-temperature in these regions is slight.

We added the relevant description in the manuscript.

Page 9 Lines 148-151: The extreme high-temperature due to SA in India are diverse among different models though the SA-driven warming is significant, which may be relevant the large uncertainty of Indian summer monsoon precipitation (Prasanna et al., 2016)

Explained Fig. 1 Probability distribution functions of surface air temperature (tas) under the high-emission scenario over Northern Africa (12–25°N, 0–35°E) and India (8–30°N, 65–80°W) during the modern (first column, 1995-2014), mid-term future (second column, 2040-2059), and long-term future (third column, 2080-2099) periods in MPI-ESM1-2-LR model.

Reference

Prasanna, V. Assessment of South Asian Summer Monsoon Simulation in CMIP5-Coupled Climate Models During the Historical Period (1850–2005). *Pure Appl. Geophys.* 173, 1379–1402 (2016). <https://doi.org/10.1007/s00024-015-1126-6>.

Berg, A., Lintner, B., Findell, K. & Giannini, A. Soil Moisture Influence on Seasonality and Large-Scale Circulation in Simulations of the West African Monsoon. *Journal of Climate* 30, 2295-2317, doi:10.1175/jcli-d-15-0877.1 (2017).

Zhou, S. *et al.* Diminishing seasonality of subtropical water availability in a warmer world dominated by soil moisture-atmosphere feedbacks. *Nat Commun* 13, 5756, doi:10.1038/s41467-022-33473-9 (2022).

Reviewer Comment

2. Lines 356-365 discussed the method to compute GHG and LA contributions to the total warming, but it is not obvious what these terms exactly represent. The authors need to present and discuss these individual contributions for the “modern” period. From Fig. 2, it seems that the LA-driven temperature difference (warming) is quite large over NA in the “modern” period. The question is: is this LA-driven large difference

realistic? – at least some comparisons with any prior studies are needed. To further ensure that the methodology is appropriate, the authors should add the results for the period of 1900-1919 (when GHG forcing is weak) - if such model simulation results are available.

Reply

According to the reviewer's suggestion, we revised and described the contribution to warming trend in more detail. Meanwhile, it is important to note that the contributions to warming trend is the warming rate by total, GHG, and SA effect, not the cumulative warming.

Page 21 Lines 328-344: Contribution to the warming trend. We calculated the warming trend ($^{\circ}\text{C}/\text{decade}$) of total, GHG, and SA effect on ground surface air temperature under high-emission scenario (SSP5-8.5) over globe (not include Antarctica), northern middle latitudes (Northern: $30\text{--}60^{\circ}\text{N}$, $180^{\circ}\text{W}\text{--}180^{\circ}\text{E}$), southern subtropical latitudes (Southern: $20\text{--}40^{\circ}\text{S}$, $180^{\circ}\text{W}\text{--}180^{\circ}\text{E}$), Europe (EUR: $40\text{--}60^{\circ}\text{N}$, $20\text{--}50^{\circ}\text{E}$), and North America (NA: $28\text{--}55^{\circ}\text{N}$, $88\text{--}110^{\circ}\text{W}$) in future projections (2015-2099). (1) The total contribution to the warming trend is obtained by calculating the trend of the surface air temperature in fully coupled experiment under high-emission scenario (SSP5-8.5), because the time series of this experiment included both GHG and SA effects; (2) the contribution of SA to the warming trend is obtained by calculating the trend of the difference of surface air temperature between fully coupled experiment (SSP5-8.5) and fixed soil moisture experiment (LFMIP-pdLC) under high-emission scenario, because the time series of this difference between the two experiments considered as SA effect; (3) the contribution of GHG to the warming trend is obtained by calculating the trend of the surface air temperature in fixed soil moisture experiment under high-emission scenario (LFMIP-pdLC), because the time series of this experiment included GHG effect and excluded SA effects.

We revised the description of the contribution to warming trend in manuscript.

Page 6 Lines 102-106: For the very high-emission pathway, the positive trends caused by SA over EUR (0.17 ± 0.08 $^{\circ}\text{C}$ pre decade) and NA (0.16 ± 0.09 $^{\circ}\text{C}$ pre decade)

correspond to $18.5 \pm 9.7\%$ and $18.8 \pm 10.0\%$, respectively, of the overall warming rate in each region (Supplementary Table 1).

Supplementary Table 1| Contributions to the warming trend ($^{\circ}\text{C}/\text{decade}$). The warming trend of total, GHG, and SA effect on ground surface air temperature under high-emission scenario (SSP5-8.5) over globe (not include Antarctica), northern middle latitudes (Northern: $30\text{--}60^{\circ}\text{N}$, $180^{\circ}\text{W}\text{--}180^{\circ}\text{E}$), southern subtropical latitudes (Southern: $20\text{--}40^{\circ}\text{S}$, $180^{\circ}\text{W}\text{--}180^{\circ}\text{E}$), Europe (EUR: $40\text{--}60^{\circ}\text{N}$, $20\text{--}50^{\circ}\text{E}$), and North America (NA: $28\text{--}55^{\circ}\text{N}$, $88\text{--}110^{\circ}\text{W}$) in future projections (2015-2099). The warming uncertainty is obtained by calculating the standard deviation of multiple models.

	Globe	Northern	Southern	EUR	NA
GHG	0.68 ± 0.13	0.77 ± 0.17	0.57 ± 0.11	0.74 ± 0.21	0.69 ± 0.17
SA	0.05 ± 0.03	0.09 ± 0.05	0.03 ± 0.02	0.17 ± 0.08	0.16 ± 0.09
Total	0.73 ± 0.13	0.86 ± 0.14	0.60 ± 0.12	0.91 ± 0.16	0.86 ± 0.15

For the phenomenon that SA-driven temperature difference (warming) is quite large over NA in the “modern” period in Fig. 2 and Fig. 1f (grey bars). the SA-driven 0.83 ± 0.51 $^{\circ}\text{C}$ warming in NA in the modern period is smaller than or equivalent to that in Berg et al. (2014) , in which SA does lead to more than 2°C over North America in 1971-2000 using the GFDL climate model (Explained Fig. 2). We have also added relevant quote and explanation in the manuscript.

Page 4 Lines 67-71: The most extreme warming occurs over central North America (NA; $28\text{--}55^{\circ}\text{N}$, $88\text{--}110^{\circ}\text{W}$) and central and eastern Europe (EUR; $40\text{--}60^{\circ}\text{N}$, $20\text{--}50^{\circ}\text{E}$) in modern period ($0.89 \pm 0.53^{\circ}\text{C}$ and $0.56 \pm 0.41^{\circ}\text{C}$), which is generally consistent with Berg et al¹⁴. The warming will up to 2.4°C in the two regions in the long-term future under the very high emission scenario (Fig. 1d).

Explained Fig. 2 (Berg et al., 2014). Difference of the distribution of daily JJA (June, July, August) 2-m mean temperature between simulations CTL experiment (fully coupled experiment) and 1A experiment (soil moisture is fixed experiment in CMIP5, such as LFMIP-pdLC experiment) over 1971-2000.

Finally, the model simulation in the period of 1900-1919 are not available because the LFMIP-pdLC experiment period is 1980-2099. As our response in the above, the SA-warming in the modern time is consistent with the previous studies, so we could believe the warming in the future is reliable to some extent.

Reference

Berg, A., et al. Impact of Soil Moisture–Atmosphere Interactions on Surface Temperature Distribution. *Journal of Climate* 27, 7976-7993, doi:10.1175/jcli-d-13-00591.1 (2014).

Reviewer Comment

3. Eq. (1) is the primary equation used to explain the mechanisms for the results, and hence the assumptions used to derive this equation should be presented. For instance, it is unclear how the downward and upward longwave radiation variations are computed.

Reply

Because the decomposition of surface air temperature changes caused by SA through surface energy balance was referenced Zeng et al. (2017), the specific

derivation process was not given in this manuscript. Nevertheless, we have added the derivation process to the Materials and Methods as suggested by the reviewer.

Pages 21-25 Lines 345-422: Decomposition of SA-driven changes in surface air temperature. The surface energy balance controls land–atmosphere interactions. There are two dominant factors in driving the changes of land-surface air temperature (T_a): first, the radiative and thermodynamic variations of land surface acts directly on land-surface temperature (T_s), the change of T_s interacts on T_a locally through radiative (for example, long wave radiation) and non-radiative (for example, sensible heat) fluxes (ΔT_a^{rad}); second, the change in atmospheric circulation (for example, advection of cold and warm air masses) acts more directly on T_a (ΔT_a^{cir}). That is:

$$\Delta T_a = \Delta T_a^{rad} + \Delta T_a^{cir} \quad (1)$$

We first estimate the change in T_s associated with the ΔSA -induced radiative and thermodynamic forcings because the land-surface energy budget is calculated at the land-surface layer. The land-surface energy balance is given by:

$$S_n + L_n = \lambda E + H + G \quad (2)$$

where S_n is the net shortwave radiation at the surface, L_n is the net longwave radiation at the surface, λ is the latent heat of vaporization, E is evapotranspiration, H is the sensible heat flux and G is the ground heat flux, which can be neglected due to its small magnitude on seasonal and longer timescales. Equation (2) can be rewritten as:

$$S_n + L_n = \lambda E + H \quad (3)$$

S_n , L_n , and H , are given as:

$$S_n = S\tau(1 - \alpha) \quad (4)$$

where S is the solar radiation flux at the top of atmosphere, τ is the atmospheric shortwave transmissivity, α is the surface albedo.

As most of atmospheric water vapor is confined near the surface, some empirical equations can provide very good estimates of downward longwave radiation worldwide using surface observations. According to the Stephan–Boltzmann law, given atmospheric air emissivity (ϵ_a), the downward longwave radiation at the land surface can be estimated roughly as

$$L_{\downarrow} = \varepsilon_a \sigma T_a^4 \quad (5)$$

where σ is the Stephan–Boltzmann constant, and σ equals $5.67 \times 10^{-8} \text{ W m}^{-2} \text{ K}^{-4}$. ε_a is atmospheric air emissivity and its variation is driven by changes in atmospheric water vapour and clouds.

The upward longwave radiation is given by

$$L_{\uparrow} = (1 - \varepsilon_s) \varepsilon_a \sigma T_a^4 + \varepsilon_s \sigma T_s^4 \quad (6)$$

where ε_s is the land-surface emissivity. Note the land-surface emissivity ε_s also changes with land cover, soil moisture and snow cover. Here we treat it as a constant of 0.95 for simplicity, as satellite-observed surface emissivity varies little over mostly vegetated surfaces and changes only slightly from 0.95 among different land covers.

Thus, the net longwave radiation over the land surface is given by

$$L_n = L_{\downarrow} - L_{\uparrow} = \varepsilon_s \sigma (\varepsilon_a T_a^4 - T_s^4) \quad (7)$$

The sensible heat flux is given by:

$$H = \frac{\rho C_d (T_s - T_a)}{r_a} \quad (8)$$

where ρ is the air density (1.21 kg m^{-3}), C_d is the specific heat of air at constant pressure ($1013 \text{ J kg}^{-1} \text{ K}^{-1}$) and r_a is the aerodynamic resistance at 2 m height.

Making use of equations (2)–(8), the surface energy balance equation is expanded into the following form:

$$S\tau(1 - \alpha) + \varepsilon_s \sigma (\varepsilon_a T_a^4 - T_s^4) = \lambda E + \rho C_d \frac{(T_s - T_a)}{r_a} \quad (9)$$

Assuming that S , λ , ρ , C_d , σ and ε_s are independent of T_s , we further differentiate the equation (9) with respect to T_s , giving the change T_s :

$$\Delta T_s = \frac{1}{f_s} \left(-S\tau\Delta\alpha - \lambda\Delta E + S(1 - \alpha)\Delta\tau + \varepsilon_s \sigma T_a^4 \Delta\varepsilon_a + \frac{\rho C_d (T_s - T_a)}{r_a^2} \Delta r_a \right) + \frac{\rho C_d / r_a + 4\varepsilon_s \sigma \varepsilon_a T_a^3}{\rho C_d / r_a + 4\varepsilon_s \sigma T_s^3} \Delta T_a \quad (10)$$

where f_s is an energy redistribution factor, given by:

$$f_s = \rho C_d / r_a + 4\varepsilon_s \sigma T_s^3 \quad (11)$$

f_s^{-1} represents the land-surface temperature sensitivity to 1 W m^{-2} radiative forcing at the land surface.

On the right-hand side of equation (10), the first term represents the land-surface temperature change due to radiative and thermodynamic forcings associated with SA-caused changes in surface albedo, evapotranspiration, shortwave transmissivity, air emissivity and aerodynamic resistance ($\Delta T_s^{rad} = (1/f) \left(-S\tau\Delta\alpha - \lambda\Delta E + S(1 - \alpha)\Delta\tau + \varepsilon_S\sigma T_a^4\Delta\varepsilon_a + \frac{\rho C_d(T_s - T_a)}{r_a^2}\Delta r_a \right)$). The second term ($\frac{\rho C_d/r_a + 4\varepsilon_S\sigma\varepsilon_a T_a^3}{\rho C_d/r_a + 4\varepsilon_S\sigma T_s^3}\Delta T_a$) quantifies the strong coupling between T_a and T_s . On the one hand, it reveals that T_s varies with T_a due to SA-induced change in air advection, such as the SA-perturbed advection of cold and warm air masses ($\Delta T_s^{cir} = \frac{\rho C_d/r_a + 4\varepsilon_S\sigma\varepsilon_a T_a^3}{\rho C_d/r_a + 4\varepsilon_S\sigma T_s^3}\Delta T_a^{cir}$). On the other hand, it also shows that the change in T_s further drives a change in T_a via the change in surface heating rate ($\Delta T_a^{rad} = \frac{\rho C_d/r_a + 4\varepsilon_S\sigma T_s^3}{\rho C_d/r_a + 4\varepsilon_S\sigma\varepsilon_a T_a^3}\Delta T_s^{rad}$). Using equations (1) and (10), surface air temperature changes caused by SA can be decomposed into the following forms:

$$\Delta T_a = \frac{1}{f} \left(-S\tau\Delta\alpha - \lambda\Delta E + S(1 - \alpha)\Delta\tau + \varepsilon_S\sigma T_a^4\Delta\varepsilon_a + \frac{\rho C_d(T_s - T_a)}{r_a^2}\Delta r_a \right) + \Delta T_a^{cir} \quad (12)$$

Where $f = \rho C_d/r_a + 4\varepsilon_S\sigma\varepsilon_a T_a^3$, f^{-1} represents the land-surface air temperature sensitivity to 1 W m^{-2} radiative forcing at the land surface.

The change in surface air temperature (ΔT_a) caused by SA is decomposed into radiative forcing terms by equation (12)³¹ and incorporating surface albedo ($-S\tau\Delta\alpha$), evapotranspiration ($-\lambda\Delta E$), shortwave transmissivity ($S(1 - \alpha)\Delta\tau$), air emissivity ($\varepsilon_S\sigma T_a^4\Delta\varepsilon_a$), aerodynamic resistance ($\frac{\rho C_d(T_s - T_a)}{r_a^2}\Delta r_a$), and residual term (ΔT_a^{cir}).

Reference

Zeng, Z. et al. Climate mitigation from vegetation biophysical feedbacks during the past three decades. *Nature Climate Change* 7, 432-436, doi:10.1038/nclimate3299 (2017).

Minor Comments:

Reviewer Comment

4. the use of the term ‘‘Snowball effect’’ in the title does not seem appropriate, as there are both positive and negative LA contributions to warming (see major comment 1).

Reply

The SA-warming occurs over most parts of the world while the SA-cooling only occurs in parts of Sahara, western Asia, and Greenland. Furthermore, the warming strength is much stronger than the cooling. So for global scale, SA play a warming role on land surface air temperature.

“Snowball” in this manuscript means the SA-warming will increase over time with greater GHG-warming in SSP5-8.5 (Fig. 2), which makes global warming accelerating—just like a snowball downhill on the snowy mountain getting bigger and bigger. Nevertheless, reviewer #3 thinks that “snowball” is easy to be confused with “snowball Earth”. Additionally, we further checked the LFMIP-pdLC experiment and found it only fixed soil moisture (do not fix snow cover).

Therefore, we revised the title to “Soil moisture-atmosphere coupling accelerates global warming” in the revised manuscript.

Reviewer Comment

5. for Table S1, provide the scenario used. Furthermore, provide the range (i.e., min and max) for each value (from the various model results).

Reply

The future scenario in Table S1 is high-emission scenario (SSP5-8.5). We have added explanation in the table title. Meanwhile, we calculated the warming trend of each model, and gave the uncertainty by calculating the standard deviation of multiple models in the table (the revised table can be seen in the reply for the major comment 2). Thanks.

Reviewer Comment

6. mention in the text how the “globe” is defined? – does it include Antarctic and Greenland?

Reply

The globe not include Antarctica, we have added this information in the revised manuscript (Page 2 Line 25; Page 4 Lines 75-76; Page 6 Lines 93-94; Page 21 Line 330). Thanks.

Reviewer Comment

7. Lines 333-337: provide more details on LFMIP-pdLC.

Reply

Following the comment, we have provided a more detailed description of LFMIP-pdLC experiment in the Introduction and Materials and Methods in the revised manuscript.

Page 3 Lines 51-57: For the Land Feedback Model Intercomparison Project with prescribed Land Conditions experiment (LFMIP-pdLC) in LS3MIP, which the soil moisture is fixed to its climatological state (the annual mean cycle for the period 1980–2014 derived from historical global climate model output, Supplementary Fig. 1). Then the SA effect can be isolated from the relative differences between fully coupled experiments (historical, the SSP1-2.6, and SSP5-8.5 experiments) and LFMIP-pdLC experiment.

Page 19 Lines 297-305: The LFMIP-pdLC experiment in LS3MIP is used to assess the impact of land-atmosphere coupling caused by soil moisture on weather and climate through fixing the soil moisture as their climatological state (the annual mean cycle for the period 1980–2014 derived from historical global climate model output, Supplementary Fig. 1). Then the SA effect can be isolated from the relative differences between fully coupled experiments (historical, the SSP1-2.6, and SSP5-8.5 experiments) and fixed soil moisture experiment (LFMIP-pdLC experiment).

Reviewer # 2:**Reviewer Comment**

The paper “Snowball effect of land-atmosphere coupling on global warming” explores the role of land-atmosphere feedbacks on projections of surface air temperature by comparing the outputs from LS3MIP experiment with the historical and future projections with SSP1-2.6 and SSP5-8.5. The results indicate that the land surface has a compounding effect on the global warming due to the feedbacks between the land and the atmosphere. Specifically, over high latitudes, the fully coupled models show a decrease in evaporation and decrease in cloud cover which cause an increase in radiation at the surface and an increase in sensible heating that led to an increase in surface air temperature and compounds the effects of greenhouse gasses.

The paper is logically organized which makes it easy to follow. The experiment design is interesting, and the results provide insights into the coupling between the land and the atmosphere and its impact on future projects of temperature. Based on this, the paper is well suited for Nature Communications and merits publication. Despite these positive aspects, the paper is not particularly clear in defining certain aspects of the work and there needs to be some discussion about the uncertainty in the models. Based on this, here are a few suggestions for improvement.

Reply

We thank the reviewer for the positive comments on the presentation and novelty of this manuscript. we have modified this article in accordance with the reviewer’s comments. Especially, we added the discussion about the uncertainty of the models in the Discussion and Conclusion, and described the experiment design and methods introduction in more detail in the materials and methods. Below please find our point-to-point response to your comments.

Reviewer Comment

1. First, there are a few places in the paper that need some clarification. For example, in lines 50-56 it provides a brief overview of the LS3MIP experiment and how the models are used to derive the results. I found this section to be confusing and it would

benefit for a little bit more detail about the LS3MIP experiment. Specifically, it uses the climatology from 1980-2014 for soil moisture, but does it also change soil temperature, snow cover and other lands surface states? What impacts does changing some aspects of the land surface model but not others have on the model simulations?

Reply

Firstly, we changed title to “Soil moisture-atmosphere coupling accelerates global warming” in the revised manuscript because the LFMIP-pdLC experiment only reflects the effect of fixed soil moisture.

In the LFMIP-pdLC experiment, changes in soil moisture over time are replaced by their climatology state (the annual mean cycle for the period 1980–2014 derived from historical global climate model output) to remove their interannual variability and long-term trends, other land surface elements (such as soil temperature, snow cover, and vegetation) are not fixed. So, the soil moisture-atmosphere coupling (SA) effect can be isolated from the differences between the fully coupled experiments (historical, the SSP1-2.6, and SSP5-8.5 experiments) and LFMIP-pdLC experiment (fixed soil moisture).

Following the comment, we have modified the description of this experiment in more detail.

Page 3 Lines 51-57: For the Land Feedback Model Intercomparison Project with prescribed Land Conditions experiment (LFMIP-pdLC) in LS3MIP, which the soil moisture is fixed to its climatological state (the annual mean cycle for the period 1980–2014 derived from historical global climate model output, Supplementary Fig. 1). Then the SA effect can be isolated from the differences between the fully coupled experiments (historical, the SSP1-2.6, and SSP5-8.5 experiments) and LFMIP-pdLC experiment.

Reviewer Comment

2. Line 104-106 is also a bit confusing. What is meant by “LA will switch from its role as an agent of internal climate variability to one in which LA is tied directly to GHG

emissions”? This statement is problematic on many levels and should be revised. It is unclear on what is meant by “agent” and how LA is “tied directly to GHG”.

Reply

The meaning of this sentence is that soil moisture-atmosphere coupling as a kind of internal climate variability is originally little affected by external forcing. However, SA will be more closely related to higher GHG emission in the future. According to the comment, we have deleted the sentence in the revised manuscript because of the plausible potential confusion for readers.

Reviewer Comment

3. Line 329 was confusing as it is unclear by what is meant by “monthly data will affect the distribution of extreme temperature probability”. Is this referencing the fact that the monthly data will not show as high of a value as daily data due to temporal averaging? If so, then it is still unclear exactly how this was resolved given that some models provide daily data and others only have monthly data.

Reply

Our presentation is not clear enough in here, because amount of monthly data is not enough compared with daily data, we are concerned that the probability distribution function (PDF) characteristics cannot be accurately captured. Meanwhile, the multi-model mean will mask the intensity distribution of extreme high-temperatures when the number of models is little. So, we used daily data to calculate the PDF and extreme high-temperature probability for each model (do not multi-model mean) separately in Fig. 3 (MPI-ESM1-2-LR model under high-emission scenario), Supplementary Fig. 4 (MPI-ESM1-2-LR model under low-emission scenario), Supplementary Fig. 5 (CMCC-ESM2), and Supplementary Fig. 6 (IPSL-CM6A-LR). And the results showed that they had similar characteristics.

We have modified the description of time-scale in more detail.

Pages 18-19 Lines 279-291: The surface air temperature probability distribution function and extreme high-temperature were calculated from daily data for each model (do not multi-model mean) separately. Because amount of monthly data is not enough

compared with daily data, we are concerned that the probability distribution function characteristics cannot be accurately captured. Meanwhile, the multi-model mean will mask the intensity distribution of extreme high-temperatures when the number of models is little. So, we used daily data to calculate the probability distribution function and extreme high-temperature probability for each model (do not multi-model mean) separately in Fig. 3 (MPI-ESM1-2-LR model under high-emission scenario), Supplementary Fig. 4 (MPI-ESM1-2-LR model under low-emission scenario), Supplementary Fig. 5 (CMCC-ESM2), and Supplementary Fig. 6 (IPSL-CM6A-LR). And the results showed that they had similar characteristics.

Reviewer Comment

4. Lines 345-351 are very confusing, and I am not completely sure I understand what is going on. Are you doing a regional scale analysis (e.g., multiple grid cells within a certain range of the center grid cell) or is it just a grid cell analysis? Using without LA interactions as the threshold temperature only works if you are comparing the differences between without LA and with LA across the three time periods (modern, mid-term and long-term). I think that is what is being done here, but it is difficult to understand.

Reply

For the description of regional scale and grid cells, we didn't describe them very clearly. The spatial distribution of extreme high-temperature (Figs. 3a-b, Supplementary Figs. 4a-b, Supplementary Figs. 5a-b, and Supplementary Figs. 6a-b) is determined by calculating the 90th percentile of surface air temperature in each spatial grid point. And for the probability distribution function of surface air temperature over NA and EUR (Figs. 3c-d, Supplementary Figs. 4c-d, Supplementary Figs. 5c-d, and Supplementary Figs. 6c-d), regional average of surface air temperature is calculated first, and then the probability distribution function of the regional average is calculated.

The reviewer's understanding about threshold temperature is correct. We have modified those description in more detail in the revised manuscript.

Pages 20-21 Lines 313-327: Extreme high-temperature. The spatial distribution of extreme high-temperature (Figs. 3a-b, Supplementary Figs. 4a-b, Supplementary Figs. 5a-b, and Supplementary Figs. 6a-b) is determined by calculating the 90th percentile of surface air temperature in each spatial grid point. And for the probability distribution function of surface air temperature over NA and EUR (Figs. 3c-d, Supplementary Figs. 4c-d, Supplementary Figs. 5c-d, and Supplementary Figs. 6c-d), regional average of surface air temperature is calculated first, and then the probability distribution function of the regional average is calculated. In order to analyze the extreme high-temperature changes driven by SA, the 90th percentile of surface air temperature in LFMIP-pdLC experiment is used as the threshold temperature in three time periods (modern, mid-term, and long-term) separately. The probability and intensity differences of surface air temperature in coupled experiment (historical, the SSP1-2.6, and SSP5-8.5 experiments) relative to the threshold temperature in LFMIP-pdLC experiment are taken as the SA effect across the three time periods (modern, mid-term, and long-term).

Reviewer Comment

5. Supplementary Fig. 1 (Lines 412-415) was not helpful and difficult to see any relationship given the monthly frequency of the data. Consider revising so that it can clearly show the difference between the two model runs (uncoupled, coupled) that you are trying to emphasize here.

Reply

The original figure is the regional average (North America), which masks a lot of differences. As suggested by the reviewer, we selected a grid in North America to redraw the figure (as shown in the below). It can be found the difference of soil moisture between the fully coupled and fixed soil moisture experiments is distinct. The original figure has been replaced by the below figure in the revised manuscript.

Supplementary Fig. 1| The multi-model mean of surface (top) and total (bottom) monthly soil moisture (kg/m^2) periods (1980–2099) for a grid in North America under high-emission scenario. Red lines represent fully coupled simulations (historical experiment in 1980-2014, and SSP5-8.5 experiment in 2015-2099); black lines represent fixed soil moisture simulations (LFMIP-pdLC in 1980-2099).

Reviewer Comment

6. Supplementary Fig. 9 (Lines 471-481) was very interesting and clearly shows the attribution of the change from the coupled and uncoupled models. Consider trying to highlight this more in the paper.

Reply

We added the relevant description about attribution of the changes in manuscript.

Thanks

Pages 14-15 Lines 224-234: Generally, the shortwave transmissivity term is larger than the evapotranspiration term for the global land in the modern period, while the effects of the two terms become equivalent in the future, especially in very high emission scenario. Over EUR and NA in the long-term future, the combined positive radiation (sum of the above five terms) dominated by evapotranspiration term and shortwave transmissivity term caused by SA will reach 33.5 ± 14.1 and $32.8 \pm 16.7 \text{ W/m}^2$ under high-emission scenario. Compared with the modern period (15.4 ± 9.9 and $22.1 \pm 9.2 \text{ W/m}^2$), the combined radiation will increase by $117.5 \pm 276.5\%$ and $48.4 \pm 53.3\%$. However, under low-emission scenario, although the combined radiation will also increase

(23.1 ± 5.9 and 25.3 ± 12.6 W/m²), they are smaller than half of those in high-emission scenario.

Reviewer Comment

7. Lastly, and most importantly, there needs to be a broader discussion about the uncertainty of LA coupling in models and how it has the potential to spin out of control and not provide realistic representation of coupling (Dirmeyer, 2013; Roundy et al., 2013; Santanello et al., 2015). Therefore, there is always the potential that what is being shown here is not necessarily an indication of what will happen in a future climate with increased GHG, but could just be coupled models snowballing into an unrealistic coupling state due to limitations of parameterization in the land surface and atmospheric models.

Dirmeyer, P. a. (2013). Characteristics of the water cycle and land–atmosphere interactions from a comprehensive reforecast and reanalysis data set: CFSv2. *Climate Dynamics*, 41(3–4), 1083–1097. <https://doi.org/10.1007/s00382-013-1866-x>.

Roundy, J. K., Ferguson, Craig R., & Wood, Eric F. (2013). Impact of land-atmospheric coupling in CFSv2 on drought prediction. *Climate Dynamics*, 1–14. <https://doi.org/10.1007/s00382-013-1982-7>.

Santanello, J. A., Roundy, J., & Dirmeyer, P. A. (2015). Quantifying the Land–Atmosphere Coupling Behavior in Modern Reanalysis Products over the U.S. Southern Great Plains. *Journal of Climate*, 28(14), 5813–5829. <https://doi.org/10.1175/JCLI-D-14-00680.1>.

Reply

There is indeed a potential to spin out of control and not provide realistic representation of coupling due to model uncertainty which limitations of parameterization in the land surface and atmospheric models in a future climate with increased GHG. As reflected in the references (recommended by the reviewer), land-surface processes such as precipitation and evapotranspiration are very important to soil moisture, while precipitation and evapotranspiration often tend to be more uncertain in

climate models. Therefore, in the fully coupled model, the deviation due to limitations of the parameterization in the land-atmosphere coupling will make the surface air temperature present greater uncertainty with the projected time.

Nevertheless, our previous study (Qiao et al., 2022, Explained Fig. 3) shows that most of the selected models have reasonable simulation on soil moisture, so the future simulation is basically reliable. Meanwhile, by analyzing the uncertainty of scenarioMIP in CMIP6, Lehner et al. (2020) showed that the uncertainty of surface air temperature projection was mainly affected by internal variability and model uncertainty (in their paper, model uncertainty refers to climate response uncertainty, which includes the uncertainty caused by limitations of the parameterization in the land-atmosphere coupling) in the early stage, but with the projected time, the proportion of internal variability and model uncertainty would decrease, while the scenario uncertainty would gradually increase and become the most important part (Explained Figs. 4 and 5 below). Scenario uncertainty means that lack of knowledge of future radiative forcing that arises primarily from unknown future greenhouse gas emissions (Lehner et al., 2020).

Explained Fig. 3 (Qiao et al., 2022) Statistics for the number of regions with TSS > 0.6 in each model compared with reanalysis datasets (ERA5 and GLDAS). The larger the model statistical results, the better the model simulation in soil moisture. Statistics results are for (top) shallow soil moisture and (bottom) deep soil moisture. The black bar represents ERA5, and the red bar represents GLDAS.

Explained Fig. 4 (Lehner et al., 2020) (f) Illustration of the sources of uncertainty in the multi-model multi-scenario mean projection. (i) Fractional contribution of individual sources (internal variability (orange), model uncertainty (blue), and scenario uncertainty (green)) to total uncertainty.

Explained Fig. 5 (Lehner et al., 2020) Fraction of variance explained by the three sources of uncertainty (first column is internal variability, second column is model uncertainty, and third column is scenario uncertainty) in projections of decadal mean

temperature changes in 2015–2024 (first row), 2045–2054 (second row) and 2085–2094 (third row) relative to 1995–2014, from CMIP6 models. Percentage numbers give the area-weighted global average value for each map.

Consequently, we added relative discussion about climate projection uncertainty in the end of this manuscript.

Pages 17-18 Lines 263-267: Additionally, although the models we used in this research have high reliability in historical simulation of soil moisture (Qiao et al., 2022), and we reduced the uncertainty by multi-model mean, it is noted that the results may have some projection uncertainty considering limitations of the parameterization in land-atmosphere coupling (Dirmeyer, 2013; Roundy et al., 2013; Santanello et al., 2015) and scenario uncertainty (Lehner et al., 2020).

Reference

- Lehner, F., Deser, C., Maher, N., Marotzke, J., Fischer, E. M., Brunner, L., Knutti, R., and Hawkins, E.: Partitioning climate projection uncertainty with multiple large ensembles and CMIP5/6, *Earth Syst. Dynam.* 11, 491–508, doi.org/10.5194/esd-11-491-2020. (2020).
- Dirmeyer, P. A. Characteristics of the water cycle and land–atmosphere interactions from a comprehensive reforecast and reanalysis data set: CFSv2. *Climate Dynamics* 41 (3–4), 1083-1097, doi.org/10.1007/s00382-013-1866-x (2013).
- Roundy, J. K., Ferguson, Craig R., & Wood, Eric F.. Impact of land-atmospheric coupling in CFSv2 on drought prediction. *Climate Dynamics*, 1–14. <https://doi.org/10.1007/s00382-013-1982-7>. (2013).
- Santanello, J. A., Roundy, J., & Dirmeyer, P. A. Quantifying the Land–Atmosphere Coupling Behavior in Modern Reanalysis Products over the U.S. Southern Great Plains. *Journal of Climate*, 28(14), 5813–5829, doi.org/10.1175/JCLI-D-14-00680.1. (2015).
- Qiao L.; Zuo Zhiyan; Xiao Dong. Evaluation of Soil Moisture in CMIP6 Simulations. *Journal of Climate*, 35(2): 779-800. (2022).

Reviewer # 3:**Reviewer Comment**

Review for “Snowball effect of land–atmosphere coupling on global warming” by Qiao et al.

Summary

Qiao et al. analyse 6 CMIP6 climate model simulations under 2 emission scenarios and the historical period in comparison with their LFMIP-pdLC simulations under LS3MIP to isolate and study the impact of LA coupling on the warming in the 21st century from the overall warming. The authors show that the impact of soil moisture and snow cover change due to GHG changes will enhance the global warming in case of the high emission scenario.

The topic of the study is relevant for understanding the contribution of LA coupling to climate change in global climate model simulations under different GHG scenarios. But it needs major revision prior to publication.

Reply

We thank the reviewer for the positive comments on the topic. We have revised the manuscript in accordance with the reviewer’s comments. Below please find our point-to-point response to the comments.

Major Comments:**Reviewer Comment**

1. The study complements the publication Zhou, S., Williams, A.P., Lintner, B.R. et al. Diminishing seasonality of subtropical water availability in a warmer world dominated by soil moisture–atmosphere feedbacks. Nat Commun 13, 5756 (2022). <https://doi.org/10.1038/s41467-022-33473-9> , however this study is not cited.

Reply

We read this paper thoroughly, which suggested that the soil moisture–atmosphere feedbacks can influence the seasonal water availability changes by changing the large-scale atmospheric circulation in a warmer climate using the same experiment (LFMIP-pdLC) in CMIP6. Specifically, during the dry season in subtropical regions and the

Amazon, declining soil moisture reduces evapotranspiration, which can increase surface water availability by modulating large-scale atmospheric circulation (soil moisture-atmosphere effect on circulation: reduced evaporative cooling caused by declining soil moisture further amplifies land surface warming, and the associated land-ocean warming contrast strengthens surface pressure differences between ocean and land, which drives anomalous ocean-to-land moisture transport and enhances moisture convergence over land). Our study focuses on the local soil moisture-atmosphere coupling (SA) effect while Zhou et al. emphasize the non-local effect of SA, which is deserve to discussed. Therefore, we have added the relevant discussion in the revised manuscript. Thanks for the information.

Page 17 Lines 259-263: Here we emphasize the local effect of SA on surface air temperature while the latest research shows that SA can also affect the large-scale atmospheric circulation under the global warming background (Zhou et al., 2022). In this view, the joint contribution of local and non-local SA effect on global land warming needs to be further investigated.

Reference

Zhou, S., Williams, A.P., Lintner, B.R. et al. Diminishing seasonality of subtropical water availability in a warmer world dominated by soil moisture–atmosphere feedbacks. *Nat Commun* 13, 5756 (2022).

Reviewer Comment

2. The title is gimmickry (“snowball effect”) and not covering the study (“effect of land–atmosphere coupling”), the study only analyses the soil moisture and snow cover effect on the global warming, not the land-atmosphere coupling effect.

Reply

“Snowball” in this manuscript means the SA-warming will increase over time with greater GHG-warming in SSP5-8.5 (Fig. 2), which makes global warming accelerating—just like a snowball downhill on the snowy mountain getting bigger and bigger.

The LFMIP-pdLC experiment originally designed to assess the land-atmosphere coupling by fixing soil moisture and snow cover in the designed stage (Bart et al., 2016). However, in the running stage, this experiment only fixed soil moisture and snow cover did not fix (Explained Fig 6, <https://wiki.c2sm.ethz.ch/LS3MIP/Tier1Experiments>). Furthermore, In order to verify whether the snow cover is fixed, we presented time series of snow cover in different models over Europe in LFMIP-pdLC experiment (Explained Fig 7), all models have the obviously interannual variabilities and decadal variations. Therefore, the experiment reflected the land-atmosphere coupling effect driven by soil moisture.

Since soil moisture effect is included in land-atmosphere coupling, but land-atmosphere coupling does not only contain soil moisture effect (such as land use, vegetation, and snow cover). Therefore, in order to describe our research more accurately, we modified “land-atmosphere coupling (LA)” to “soil moisture-atmosphere coupling (SA)” in the title and main text.

So, we change the title: “Soil moisture-atmosphere coupling accelerates global warming” in the revised manuscript.

- Land-Hist:** 1901-2014 land-only run, at the native resolution of the land model in the CMIP6 coupled runs, using GSWP3 forcing;
 - These forcing data are currently available here and will soon be published in input4MIPs.
- LFMIP-rmLC + SST:** 1980-2100 simulation in land-atmosphere configuration;
 - SST and sea ice from the first ensemble member of the historical and the ssp126 and ssp585 scenario runs of the corresponding coupled model;
 - Soil humidity prescribed as 30-year monthly running means taken from these runs.
 - The reference simulations for this run are the historical and the ssp126 and ssp585 scenario runs.
 - Only one ensemble member is Tier 1. Additional ensemble members (suggested: 4 additional runs) are Tier 2.
 - ssp126 runs are to be tagged rXIXpXF1; ssp585 runs are rXIXpXF2.
- LFMIP-pdLC + SST:** 1980-2100 simulation in land-atmosphere configuration;
 - SST and sea ice from the first ensemble member of the historical and the ssp126 and ssp585 scenario runs of the corresponding coupled model;
 - Soil humidity prescribed as the 1980-2014 climatological monthly mean of the first ensemble member of the historical run.** (highlighted in pink)
 - The reference simulations for this run are the historical and ssp126 and ssp585 scenario runs.
 - Only one ensemble member is Tier 1. Additional ensemble members (suggested: 4 additional runs) are Tier 2.
 - ssp126 runs are to be tagged rXIXpXF1; ssp585 runs are rXIXpXF2.

Explained Fig 6 The introduction to the LS3MIP and LFMIP-pdLC experiment. (<https://wiki.c2sm.ethz.ch/LS3MIP/Tier1Experiments>)

Explained Fig. 7 Monthly time series of snow cover area percentage (%) in LFMIP-pdLC experiment over Europe (40–60°N, 20–50°E).

Reference

Bart van den Hurk, et al. LS3MIP (v1.0) contribution to CMIP6: the Land Surface, Snow and Soil moisture Model Intercomparison Project – aims, setup and expected outcome. *Geoscientific Model Development* 9, 2809-2832, doi:10.5194/gmd-9-2809-2016 (2016)

Reviewer Comment

3. This study needs to account at least for the root zone soil moisture. Showing the 10 cm soil moisture in Fig 4 is not sufficient for the goal of this study, it needs to be at least 1m. E.g. Fig. 4 shows only about 1-2 kg/m² in soil moisture change 100 years, i.e. 1-2 mm in 10 cm of soil, i.e. 2.5-5% of the porosity. Further Evapotranspiration uses the root zone, not the top 10 cm.

Reply

We chose shallow soil moisture (10 cm) for analysis because surface soil moisture has a closer relationship with evapotranspiration and can directly interact with the atmosphere.

We also investigated the root zone soil moisture (100 cm) as suggested by the reviewer (Supplementary Fig. 9 in revised manuscript). The changes of root zone soil moisture and its positive correlation with evapotranspiration are similar to those in the shallow soil moisture. We also added some description in manuscript.

Page 14 Lines 209-211: These characteristics are not only shown in the surface soil moisture, which is most directly related to evapotranspiration, but also in the root zone soil moisture (Supplementary Fig. 9).

As for the shallow soil moisture change is only about 1-2 kg/m² mentioned by the reviewer, we can find that the seasonal fluctuation in surface soil moisture over NA is about 5 kg/m² (Supplementary Fig. 1). In this view, the change is not small.

Finally, evapotranspiration in this manuscript refers to evapotranspiration of land surface, including evaporation of land surface and transpiration of vegetation, which is also a variable used in most land surface processes.

Supplementary Fig. 9| Same as Fig. 4, but for the soil moisture depth is 100 cm.

Supplementary Fig. 1| The multi-model mean of monthly surface soil moisture (10 cm, kg/m²) periods (1980–2099) for a grid in North America under high-emission scenario. Red line represents fully coupled simulations (historical experiment in 1980-2014, and SSP5-8.5 experiment in 2015-2099); black line represents fixed soil moisture simulations (LFMIP-pdLC in 1980-2099).

Reviewer Comment

4. It is necessary to distinguish regions with latent heat flux limited by soil moisture and those limited by radiation, because they will result in different land atmosphere coupling paths. Only under soil moisture limitation the soil moisture impacts the temperature. This is very well seen in the results by Qiao et al. but not discussed. Include this in the analyses.

Reply

According to our study, the SA effect makes surface air temperature increase in most parts of the world (Fig. 1), which is mainly due to the decrease of evapotranspiration caused by SA. The decreasing evapotranspiration will enlarge the solar shortwave reaching the ground and increase sensible heat flux (Fig. 4 and Fig. 5b in the revised manuscript). Nevertheless, there is still a large regional difference in warming strength caused by SA. The difference of the warming strength is mainly depending on whether soil moisture anomalies can significantly change the allocation of latent heat flux and sensible heat flux via evapotranspiration. We can find that soil moisture can decrease latent heat flux significantly in the Northern mid-latitude and Southern subtropical, especially in EUR and NA (Figs. 5c-d in the revised manuscript), in which the warming degree are most.

According to the reviewer's suggestion, we added analyze of different region with latent heat flux limited by soil moisture and those limited by radiation.

Page 11 Lines 174-177: The increasing sensible heat flux caused by the joint enhanced shortwave radiation and reduced latent heat flux conduce to the nonlinear warming under severe GHG emission. Those phenomena are the strongest in the Northern mid-latitude and Southern subtropical, especially in EUR and NA, where decreasing soil moisture can significantly change more surface energy allocation from latent heat flux to sensible heat flux via decreasing evapotranspiration.

In Detail:

Reviewer Comment

1. Snowball in the frame of climate research is associated with the “Snowball Earth” and therefore the “snowball effect” even though meaning something different may lead to the wrong attention. Please change the title to fit the study (see summary) and not to be gimmickry.

Reply

Please see our response to Major Comments 2, and we change the title as follow: “Soil moisture-atmosphere coupling accelerates global warming”.

Reviewer Comment

2. Lines 44-48: Something is missing here. It is not clear how many models etc without jumping to the methods section.

Reply

Thanks, the specific models used in this manuscript are added in the main text.

Page 3 Lines 44-46: In order to analyze the SA effect on global warming, we employed six global climate models (CESM2, CMCC-ESM2, EC-Earth3, IPSL-CM6A-LR, MIROC6, and MPI-ESM1-2-LR)...

Reviewer Comment

3. Line 51: LFMIP simulations are not “uncoupled”. It is important to name here LFMIP so nobody is confused with uncoupled LMIP (Bart Van den Hurk et al. Geoscientific Model Development, DOI 10.5194/gmd-9-2809-2016).

Reply

We have revised the description about LFMIP-pdLC experiment. Meanwhile, we used “fixed soil moisture experiment / LFMIP-pdLC experiment” instead of “uncouple experiment” in the revised manuscript, thanks.

Page 3 Lines 51-57: For the Land Feedback Model Intercomparison Project with prescribed Land Conditions experiment (LFMIP-pdLC) in LS3MIP, which the soil moisture is fixed to its climatological state (the annual mean cycle for the period 1980–2014 derived from historical global climate model output, Supplementary Fig. 1). Then the SA effect can be isolated from the relative differences between fully coupled experiments (historical, the SSP1-2.6, and SSP5-8.5 experiments) and LFMIP-pdLC experiment.

Reviewer Comment

4. Change lines 55-56, e.g.: “For the LFMIP experiment, land-surface forcing comprises the annual mean cycle of soil moisture and snow cover for the period 1980–2014 derived from historical global climate model output by fixing their climatological state to remove long-term trends and inter-annual variability (Supplementary Fig. 1)”. This is because not the land surface state but the soil moisture and snow cover are fixed, which is essential for the analyses because this means that the surface energy fluxes and land surface temperature can change in LFMIP.

Reply

We have revised the description about LFMIP-pdLC experiment in manuscript. The details of the modification can be seen in the previous reply and reply of Major Comment #2.

Reviewer Comment

5. Lines 55: the LA effect cannot be isolated , only that of soil moisture and snow cover.

Reply

We have revised the description about LFMIP-pdLC experiment in manuscript. The details of the changes can be seen in our response to Major Comment #2 and Detail Comment #3.

Reviewer Comment

6. Line 57-58: Why do you use 20year climatologic periods instead of 30year periods? You are looking at soil moisture and ET changes, not only temperature. See e.g. Liersch et al 2020 Environ. Res. Lett. 15 104014 DOI 10.1088/1748-9326/aba3d7 and Hawkins, E., & Sutton, R. (2016). Connecting Climate Model Projections of Global Temperature Change with the Real World, Bulletin of the American Meteorological Society, 97(6), 963-980. Retrieved Feb 28, 2023, from <https://journals.ametsoc.org/view/journals/bams/97/6/bams-d-14-00154.1.xml>.

Reply

Although a longer reference period may increase the robustness of the conclusion and better represent the climate state, there is no perfect choice of reference period as Hawkins and Sutton (2016) pointed out. In this study, we analyzed the same period as the Sixth Assessment Report (AR6) of IPCC Working Group I. In AR6, the experts analyzed climate change mainly for three time periods: the modern (1995-2014), the mid-term (2040-2059), and the long-term (2080-2099). Because most researches on climate change are based on parameters set by AR6, which makes it convenient to compare and discuss with other studies.

Reference

Hawkins, E., & Sutton, R. (2016). Connecting Climate Model Projections of Global Temperature Change with the Real World, Bulletin of the American Meteorological Society, 97(6), 963-980.

Reviewer Comment

7. Line 60/61: Change to: Under each emission scenario investigated here, LA due to soil moisture and snow cover amplifies global warming over much of Earth's land surface (Fig. 1).

Reply

Thanks, we have modified this sentence as reviewer's suggestion in the revised manuscript (Page 4 Lines 62-63: Under each emission scenario investigated here, soil moisture-atmosphere coupling (SA) amplifies global warming over much of Earth's land surface (Fig. 1)).

Reviewer Comment

8. Lines 63-68: Mention here the reasons for those regions. It is not surprising to find the amplification in regions which will become increasingly soil moisture limited in evapotranspiration.

Reply

We have discussed this comment in Major Comment 4, and we added some description in here. The details of the changes can be seen in our response to Major Comment #4.

9. SAT: Use the ESGF abbreviation tas, not SAT, which is rather uncommon.

Reply

We have modified the abbreviation in manuscript, thanks.

10. Line 337: Delete „Such as.

Reply

We have modified this sentence according to the reviewer's suggestion.

11. All Figs with “LA induced” differences (e.g. Fig. 4, Extended Figs. 7,8,10): Name that these are CMIP6 and LFMIP difference plots (as in Fig. 1).

Reply

We have modified this description according to the reviewer's suggestion. We used “soil moisture-atmosphere coupling (SA)” instead of “LA”, and “fixed soil moisture experiment / LFMIP-pdLC experiment” instead of “uncouple experiment” in the revised manuscript.

12. Line 34: Too many references, some are doubling the statement. References 2,3,6 and 7 are superficial.

Reply

We have deleted those references. Thanks.

13. Reference 8 has the wrong first author. It should be Berg et al.

Reply

Thanks, we have revised this reference.

14. Line 41: Decide to cite reference 22 or 24 because they are not providing any extra information needed for your study.

Reply

We have deleted those references. Thanks.

15. Figure 8: Why is figure 8 only in the Supplementary when your title is about land-atmosphere coupling?

Reply

We have put the key information in the Supplementary Fig. 8 into main text as Fig. 5 in the revised manuscript as suggested by reviewer. A more detailed description has also been added in the revised manuscript.

Pages 11-12 Lines 166-183: GHG-driven warming is projected to dry the soil column²⁶ (Figs. 4a–b and Supplementary Fig. 7), thereby reducing evapotranspiration, allowing the ground surface to receive more solar shortwave radiation (longwave radiation is much weaker) through a reduction in cloud cover (Figs. 5a-b and Supplementary Fig. 8). Meanwhile, decreasing evapotranspiration could increase

sensible heat flux from the land surface to the low-level atmosphere via decreasing the latent heat flux (Figs. 5c-d). The increasing sensible heat flux caused by the joint enhanced shortwave radiation and reduced latent heat flux conduce to the nonlinear warming under severe GHG emission. Those phenomena are the strongest in the Northern mid-latitude and Southern subtropical, especially in EUR and NA, where decreasing soil moisture can significantly change more surface energy allocation from latent heat flux to sensible heat flux via decreasing evapotranspiration. On the contrary, there is a slight cooling effect caused by SA in a few regions, such as Sahara and Arabian Peninsula. The evapotranspiration change over those regions is little, while the negative cloud cover and positive shortwave radiation are generally significant, which suggests that the local evapotranspiration associated with soil moisture is not the primary factor dominating the surface energy (Supplementary Fig. 8 and Fig 5c-d). The impact of SA on surface energy over those regions may be due to the non-local effect and related to the large-scale circulation.

Fig. 5| Spatial distributions of land surface meteorological elements caused by SA in modern (1995-2014), mid-term (2040-2059), and long-term future (2080-2099) periods under the very high-emission scenario (SSP5-8.5). a is sensible heat flux (W/m^2), b is latent heat flux (W/m^2), c is total cloud cover (%), and d is surface-received shortwave radiation (W/m^2). Black dots signify agreement between the sign of change and the multi-model mean in at least five of the six (or four of five) CMIP6 models.

16. How is the albedo changing? Due to snow and moisture or also due to dynamic vegetation in the models?

Reply

In the six climate models, CESM2, CMCC-ESM2, IPSL-CM6A-LR, and MPI-ESM1-2-LR models include simulated dynamic vegetation, and EC-Earth3 and MIROC6 models do not include simulated dynamic vegetation. So, in the former four climate models, the surface albedo is also related to dynamic vegetation caused by soil moisture. The albedo change has been calculated by formula 12 (Materials and Methods) in Supplementary Figs. 10 and 11a, which mainly reflects the influence of albedo changing caused by soil moisture on surface air temperature. We can find, no matter with or without dynamic vegetation, the effect of albedo changing on surface air temperature is very weak.

17. Do the models include simulated dynamic vegetation (i.e. LAI growth and root growth due to atmospheric and soil moisture conditions)? Add this information.

Reply

For the six models used in this manuscript, CESM2, CMCC-ESM2, IPSL-CM6A-LR, and MPI-ESM1-2-LR models include simulated dynamic vegetation, and EC-Earth3 and MIROC6 models do not include simulated dynamic vegetation. We have added relevant information in the revised manuscript.

Page 19 Lines 291-295: Six global climate models in CMIP6 (CESM2, CMCC-ESM2, EC-Earth3, IPSL-CM6A-LR, MIROC6, and MPI-ESM1-2-LR) are used to analyze the impact of SA on surface air temperature, which CESM2, CMCC-ESM2, IPSL-CM6A-LR, and MPI-ESM1-2-LR models include simulated dynamic vegetation, and EC-Earth3 and MIROC6 models do not include.

REVIEWERS' COMMENTS

Reviewer #1 (Remarks to the Author):

The authors have satisfactorily addressed my concerns. The manuscript can be accepted after addressing the following minor comments.

(1) In the derivation of Eq. (10), the change of latent heat flux (LH) is used directly, while the change of sensible heat flux is written as a function of the change of $(T_s - T_a)$. In contrast, in the popular Penman-Monteith equation derivation, the change of LH is also written as a function of the change of T_a . please add a few sentences to justify the decision (of using the change of LH directly in Eq. (10) without explicitly considering its dependence on the change of T_a).

(2) In the last term of Eq. (10) and in lines 404, 407, and 409, replace " $\rho C_d/r_a + 4 \epsilon_s \sigma T_s^3$ " by f_s for simplicity, as f_s is clearly defined in Eq. (11).

(3) The language needs to be significantly improved. Just taking Abstract as an example: "...warming globe (not include ...of SA on climate changeand make a nonlinear ..." should probably be: "...warms globe (excluding ...of SA to climate changeand play a nonlinear ..."

Reviewer #2 (Remarks to the Author):

Overall, the authors did a good job of addressing my concerns. One area of minor concern is that the authors did a good job of addressing my concern related to "uncertainty of LA coupling in models" in the response document but the amount of discussion that made it into the paper was minimal (Lines 263-267). Specifically, I think the discussion would be improved if the authors added one or two more lines to this discussion arguing why they think the results are still relevant even with these uncertainties (I think they are still relevant and useful). The authors can get these additional lines by summarizing the arguments made in the response document.

Reviewer #3 (Remarks to the Author):

Dear Liang Qiao et al.,
thanks for the very good revision of your manuscript. You answered all my comments satisfyingly. So I recommend its publication.
The following minor corrections are required:
It needs somebody checking the spelling and language, sometimes verbs (e.g. line 46, line 70) or letters are missing. Fig. 3c has overlapping x-axes.
I am looking forward to see the published manuscript.

Responses to NCOMMS-22-53760B

Reviewer #1:

Reviewer Comment

The authors have satisfactorily addressed my concerns. The manuscript can be accepted after addressing the following minor comments.

Reply

We thank the reviewer for the positive comments and recognition of the conclusions. Below please find our point-to-point response to the comments.

Reviewer Comment

(1) In the derivation of Eq. (10), the change of latent heat flux (LH) is used directly, while the change of sensible heat flux is written as a function of the change of $(T_s - T_a)$. In contrast, in the popular Penman-Monteith equation derivation, the change of LH is also written as a function of the change of T_a . please add a few sentences to justify the decision (of using the change of LH directly in Eq. (10) without explicitly considering its dependence on the change of T_a).

Reply

T_a does affect latent heat, but in water-restricted areas such as arid and semi-arid regions, the change in LH is more determined by soil moisture, so LH is not written as a function of the change of T_a .

We have added explanation in the revised manuscript according to the reviewer's suggestion.

Page 23 Lines 388-390: Because in water-restricted areas such as arid and semi-arid regions, the change of latent heat flux is more determined by soil moisture, so latent heat flux is not written as a function of the change of T_a .

Reviewer Comment

(2) In the last term of Eq. (10) and in lines 404, 407, and 409, replace “ $\rho C_a/r_a + 4\varepsilon_s\sigma T_s^3$ ” by f_s for simplicity, as f_s is clearly defined in Eq. (11).

Reply

Thanks, we have used the abbreviation of f_s in relevant equations according to the reviewer's suggestion (Page 24 Lines 406, 409, 411).

(3) The language needs to be significantly improved. Just taking Abstract as an example:

“...warming globe (not include ...of SA on climate changeand make a nonlinear ...”, should probably be: “...warms globe (excluding ...of SA to climate changeand play a nonlinear ...”

Reply

Thanks, we have checked and modified the grammar in the revised manuscript.

Such as:

Pages 2 Lines 20: “predicts” to “projects”

Pages 2 Lines 21: “warming” to “warms”

Pages 2 Lines 25: “not include” to “excluding”

Pages 2 Lines 27: “on” to “to”

Pages 2 Lines 28: “make a non-linear warming role” to “play a non-linear warming component role”

Pages 2 Lines 32: “have” to “has”

Pages 2 Lines 33: “the rate of warming varies” to “the rates of warming vary”

Pages 2 Lines 34: “Previous research has” to “Previous researches have”

Pages 2 Lines 36: “the fluxes of mass” to “mass fluxes”

Pages 3 Lines 49: added “— to isolate the climatic impact of SA”

.....

Reviewer #2:

Reviewer Comment

Overall, the authors did a good job of addressing my concerns. One area of minor concern is that the authors did a good job of addressing my concern related to “uncertainty of LA coupling in models” in the response document but the amount of discussion that made it into the paper was minimal (Lines 263-267). Specifically, I think

the discussion would be improved if the authors added one or two more lines to this discussion arguing why they think the results are still relevant even with these uncertainties (I think they are still relevant and useful). The authors can get these additional lines by summarizing the arguments made in the response document.

Reply

We thank the reviewer for the positive comments and recognition of the conclusions. We have modified and added relevant discussion about uncertainties in the revised manuscript.

Page 17 Lines 263-270: Finally, the results may have some projection uncertainty considering the limitations of the parameterization in SA (Dirmeyer, 2013; Roundy et al., 2013; Santanello et al., 2015) and scenario uncertainty due to the lack of knowledge of future radiative forcing (Lehner et al., 2020). Nevertheless, the models we used in this research have high reliability in the historical simulation of soil moisture (Qiao et al., 2022), and we reduced the uncertainty by multi-model mean. Meanwhile, different single models have similar results for the acceleration in SA-driven warming in the future projection. So, the conclusion that SA amplifies greenhouse gas-driven global warming is relatively reliable and robust.

Reviewer #3:

Reviewer Comment

Dear Liang Qiao et al., thanks for the very good revision of your manuscript. You answered all my comments satisfyingly. So I recommend its publication. The following minor corrections are required: It needs somebody checking the spelling and language, sometimes verbs (e.g. line 46, line 70) or letters are missing. Fig. 3c has overlapping x-axes. I am looking forward to see the published manuscript.

Reply

We thank the reviewer for the positive comments and recognition of the conclusions. We adjusted the x-axes in Fig. 3c, and we have checked and modified the spelling and grammar in the revised manuscript. Such as:

Pages 2 Lines 20: “predicts” to “projects”

Pages 2 Lines 21: “warming” to “warms”

Pages 2 Lines 25: “not include” to “excluding”

Pages 2 Lines 27: “on” to “to”

Pages 2 Lines 28: “make a non-linear warming role” to “play a non-linear warming component role”

Pages 2 Lines 32: “have” to “has”

Pages 2 Lines 33: “the rate of warming varies” to “the rates of warming vary”

Pages 2 Lines 34: “Previous research has” to “Previous researches have”

Pages 2 Lines 36: “the fluxes of mass” to “mass fluxes”

Pages 3 Lines 49: added “— to isolate the climatic impact of SA”

Pages 3 Lines 51: deleted “each”

Pages 3 Lines 52: “For” to “in”

Pages 3 Lines 53: “in” to “from”

Pages 3 Lines 54: deleted “which”, “the annual mean cycle”

Pages 3 Lines 54: added “which is”

Pages 3 Lines 58: “used the three climatology periods during” to “consider three time horizon periods”

Pages 4 Lines 68: deleted “the”

Pages 4 Lines 73: “will” to “will be”

.....